



# Wake properties and power output of very large wind farms for different meteorological conditions and turbine spacings: A large-eddy simulation case study for the German Bight

Oliver Maas[1] and Siegfried Raasch[1]

[1]Institute of Meteorology and Climatology, Leibniz University Hannover, Hannover, Germany

**Correspondence:** Oliver Maas (maas@muk.uni-hannover.de)

**Abstract.** Germany's expansion target for offshore wind power capacity of 40 GW by the year 2040 can only be reached if large portions of the Exclusive Economic Zone in the German Bight are equipped with wind farms. Because these wind farm clusters will be much larger than existing wind farms, it is unknown how they affect the boundary layer flow and how much power they will produce. The objective of this large-eddy-simulation study is to investigate the wake properties and the power output of very large potential wind farms in the German Bight for different turbine spacings, stabilities and boundary layer heights. The results show that very large wind farms cause flow effects that small wind farms do not. These effects include, but are not limited to, inversion layer displacement, counterclockwise flow deflection inside the boundary layer and clockwise flow deflection above the boundary layer. Wakes of very large wind farms are longer for shallower boundary layers and smaller turbine spacings, reaching values of more than 100 km. The wake in terms of turbulence intensity is approximately 20 km long, where longer wakes occur for convective boundary layers and shorter wakes for stable boundary layers. Very large wind farms in a shallow, stable boundary layer can excite gravity waves in the overlying free atmosphere, resulting in significant flow blockage. The power output of very large wind farms is higher for thicker boundary layers, because thick boundary layers contain more kinetic energy than thin boundary layers. The power density of the energy input by the geostrophic pressure gradient limits the power output of very large wind farms. Because this power density is very low (approximately $2\,\mathrm{W\,m^{-2}}$), the installed power density of very large wind farms should be small to achieve a good wind farm efficiency.

## 1 Introduction

At present, the global installed wind power capacity from offshore wind farms is increasing rapidly. According to the expansion targets of the current leading offshore wind markets (the United Kingdom, Germany and China), the offshore wind power capacity will be subject to significant growth over the next decades. The German expansion target for offshore wind power capacity is 40 GW by the year 2040, which is more than the global installed offshore wind power capacity of 32.5 GW in the year 2020 (WindSeeG, 2020; Herzig, 2020). The otherwise undisturbed flow at offshore sites will be increasingly modified by wind farms, affecting the wind farm power output but also the meteorological conditions in the wake. For wind farms with a state-of-the-art size of approximately 100 turbines and a length of approximately 5 km, these effects have been extensively investigated experimentally and numerically and are generally well understood.



However, the size of future wind farms or clusters of wind farms will be one or two orders of magnitude larger than today's (cf. Fig. 1). Because no wind farms of this size exist currently, new insights into the behaviour of the flow through wind farms and the resulting power output can only be provided by simulations. The most accurate method that resolves all the relevant processes such as the turbulent momentum and heat transport that is still computationally feasible is large-eddy simulation. In recent years many large-eddy simulations of wind farm flows have been carried out. A comprehensive review can be found in Porté-Agel et al. (2020). Some of the investigations consisted of an infinite wind farm setup with cyclic boundary conditions in the streamwise and crosswise direction (e.g. Lu and Porté-Agel (2011), Calaf et al. (2011) and Johnstone and Coleman (2012)). With these methods, the limiting case of an infinite wind farm can be investigated at relatively low computational cost due to the small domain size. Johnstone and Coleman (2012) used this method to compare a neutral boundary layer flow with and without wind turbines. The wind turbines increased the boundary layer height and the ageostrophic wind component inside the boundary layer, which lead to a higher energy input by the pressure gradient. Simple 1D-models for the wind speed profile inside and above an infinite wind farm have been developed by e.g. Frandsen (1992), Calaf et al. (2010) and Abkar and Porté-Agel (2013).

Some authors used a semi-infinite wind farm setup with cyclic boundary conditions only in the crosswise direction (Stevens et al., 2016; Allaerts and Meyers, 2017; Wu and Porté-Agel, 2017). Allaerts and Meyers (2017) simulated a 15 km long wind farm in a conventionally neutral boundary layer (CNBL) with different heights. In the shallow boundary layer cases, the wind-farm-induced flow deceleration led to upward displacement of the inversion layer which triggered stationary gravity waves in the free atmosphere. These gravity waves can impose favorable and unfavorable streamwise pressure gradients upstream, inside and downstream of the wind farm, which can result in significant flow acceleration or deceleration.

Large-eddy simulations of existing wind farms have been carried out, e.g. the wind farms Horns Rev with eighty 2 MW turbines (Porté-Agel et al., 2013; Wu and Porté-Agel, 2015), alpha ventus with twelve 5 MW turbines and EnBW Baltic 1 with twenty-one 2.3 MW turbines (Witha et al., 2014).

To date, there have been no studies of wind farms of finite size with variable meteorological conditions, nor have spatial and energy scales of future wind farms (on the order of 100 km and 10 GW) been investigated. With this study we want to fill this gap, by performing large-eddy simulations of very large, finite size wind farms for different stabilities, turbine spacings and boundary layer heights. We provide new insights into the wake properties and power output of very large wind farms and how these depend on the varied parameters. Specifically we want to answer these questions:

1. How is the flow inside and above the boundary layer affected by very large wind farms?

2. How long is the wake in terms of speed deficit and turbulence intensity?

3. What physical processes drive the wake recovery?

4. How much power output or power density can be expected for very large wind farms?

5. What effect does the turbine spacing and the boundary layer height have on questions 1-4?

**Figure 1.** Existing wind farms and priority areas for future wind farms in the German Exclusive Economic Zone in the German Bight. Map is based on data that are publicly available at https://www.geoseaportal.de.

Instead of using an idealized wind farm shape, we investigate a potential future wind farm scenario in the German Bight, which is shown in Fig. 1. The scenario assumes that all priority areas for future wind farms are equipped with 15 MW wind turbines. This results in a total number of up to 2088 wind turbines with a total wind farm capacity of up to 31 GW. More than

60 7 billion grid points are required to fill the large domain with a turbine wake resolving grid. To our knowledge, this large-eddy simulation case study exceeds other studies in terms of wind farm area and total wind turbine number by at least one order of magnitude.

The numerical model, setup and boundary conditions are described in section 2. The simulation results regarding the wake properties and the power output are shown and discussed in section 3. Section 4 concludes and discusses the results of the

65 study.



## 2 METHODS

### 2.1 Numerical model

The simulations were performed with the **PA**rallelized **L**arge-eddy simulation **M**odel **PALM** (Maronga et al., 2020), which is developed at the Institute of Meteorology and Climatology of Leibniz Universität Hannover, Germany. Several wind farm flow investigations have been successfully conducted with this code in the past (e.g., Witha et al., 2014; Dörenkämper et al., 2015). PALM solves the non-hydrostatic, incompressible Navier-Stokes equations in Boussinesq-approximated form, spatially filtered over a grid volume. The equations for the conservation of mass, momentum and internal energy then read as:

$$\frac{\partial u_j}{\partial x_j} = 0, \tag{1}$$

$$\frac{\partial u_i}{\partial t} = -\frac{\partial u_i u_j}{\partial x_j} - \epsilon_{ijk} f_j u_k + \epsilon_{i3j} f_3 u_{g,j} - \frac{1}{\rho_0} \frac{\partial \pi^*}{\partial x_i} + g \frac{\theta_v - \langle \theta_v \rangle}{\langle \theta_v \rangle} \delta_{i3} - \frac{\partial}{\partial x_j} \left( \overline{u_i'' u_j''} - \frac{2}{3} e \delta_{ij} \right), \tag{2}$$

$$\frac{\partial \theta}{\partial t} = -\frac{\partial u_j \theta}{\partial x_j} - \frac{\partial}{\partial x_j} \left( \overline{u_i'' \theta''} \right), \tag{3}$$

where an overbar indicates filtered quantities and a double prime indicates subgrid-scale (SGS) quantities, $i, j, k \in \{1, 2, 3\}$, $u_i$, $u_j$, $u_k$ are the velocity components in the respective directions $(x_i, x_j, x_k)$, $t$ is time, $f_i = (0, 2\Omega\cos(\phi), 2\Omega\sin(\phi))$ is the Coriolis parameter with the Earth's angular velocity $\Omega = 0.729 \times 10^{-4} \text{rads}^{-1}$ and the geographical latitude $\phi$. The geostrophic wind speed components are $u_{g,j}$ and the basic state density of dry air is $\rho_0$. The modified perturbation pressure is $\pi^* = p^* + \frac{2}{3}\rho_0 e$, where $p^*$ is the perturbation pressure and $e = \frac{1}{2}\overline{u_i'' u_i''}$ is the SGS turbulence kinetic energy. The gravitational acceleration is $g = 9.81 \text{ ms}^{-2}$ and $\delta$ is the Kronecker delta.

The SGS model uses a 1.5-order closure according to Deardorff (1980), modified by Moeng and Wyngaard (1988) and Saiki et al. (2000). Recently, the modified version of Dai et al. (2021) has been implemented in PALM, which allows for coarser grid spacings in stable boundary layers due to reduced grid spacing sensitivity. This modified version is used for the simulation of wind farms in a stable boundary layer.

The following features of PALM are relevant for the performed simulations. It is possible to prescribe a surface heating or cooling rate, instead of prescribing a surface heat flux. Stable boundary layers can also be generated by imitating warm air advection by using a large-scale forcing. Convective boundary layer growth can be compensated by applying a large-scale subsidence to the potential temperature field. A Rayleigh damping layer can be used in order to avoid gravity wave reflection at the top of the domain.

The wind turbines are represented by an advanced actuator disc model (ADM) that acts as an axial momentum sink and an angular momentum source (inducing wake rotation). The ADM is described in detail by Steinfeld et al. (2016). The disc element forces are distributed to the neighbouring grid points by a smearing kernel, which causes a power overestimation of 12.5 % for a smearing kernel radius of $1\Delta x$. The wind turbine power output is corrected for the power overestimation before entering the wind farm power output analysis



## 2.2 Case selection

To produce meaningful and relevant results, the simulations should represent the most common meteorological conditions in
the German Bight. A climatology with frequency distributions of wind speed, wind direction, boundary layer (BL) height and
stability information extracted from the COSMO-REA6 reanalysis dataset can be found in Appendix A. The analysis was
provided by Thomas Spangehl (German Weather Service) and it is based on hourly data of a 24-year period (1995-2018) at
$54° \, 30'$ N, $6° \, 00'$ E, which is located inside Zone 3 (cf. Fig. 1). Wind speed and direction are evaluated at 178 m height, which
is the closest COSMO model level to the hub height of 150 m of the wind turbine used in the simulations.

Due to the high computational cost per simulation, only a limited number of simulations were carried out. This study consists
of five simulations with varying stability, turbine spacing and BL height. An overview is given in Table 1. Two cases with a
neutral boundary layer (NBL), two cases with a convective boundary layer (CBL) and one case with a stable boundary layer
(SBL) are simulated.

**Table 1.** Overview of simulated cases with boundary layer height $h$, turbine spacing $s$, surface heating rate $\dot{\theta}_0$ or large-scale forcing advection
tendency $\dot{\theta}_{lsf}$ in case SBL-300-7D, surface heat flux $Q_{H,0}$, Monin-Obukhov length $L$, subsidence velocity $w_{sub}$, geostrophic wind speed $G$
and direction $\alpha$, length and width of the precursor domain $L_{x,pre}$ and $L_{y,pre}$, domain height $L_z$, number of vertical grid points $n_z$, stretch
level $z_s$, Rayleigh damping level $z_{rd}$.

| Case | $h$ | $s$ | $\dot{\theta}_0 / \dot{\theta}_{lsf}$ | $Q_{H,0}$ | $L$ | $w_{sub}$ | $G$ | $\alpha$ | $L_{x,pre}$ | $L_{y,pre}$ | $L_z$ | $n_z$ | $z_s$ | $z_{rd}$ |
|---|---|---|---|---|---|---|---|---|---|---|---|---|---|---|
| unit | m | - | K h$^{-1}$ | K m s$^{-1}$ | m | mm s$^{-1}$ | m s$^{-1}$ | ° | km | km | m | - | m | m |
| NBL-700-7D | 700 | 7D | 0 | 0 | $\pm\infty$ | 0 | 10.77 | 8.9 | 5.76 | 4.80 | 2042 | 88 | 1500 | 1600 |
| NBL-700-5D | 700 | 5D | 0 | 0 | $\pm\infty$ | 0 | 10.77 | 8.9 | 5.76 | 4.80 | 2042 | 88 | 1500 | 1600 |
| CBL-700-7D | 700 | 7D | 0.05 | +0.007 | −420 | 3.968 | 10.19 | 9.5 | 7.68 | 3.84 | 2042 | 88 | 1500 | 1600 |
| CBL-1400-7D | 1400 | 7D | 0.025 | +0.008 | −390 | 1.984 | 10.13 | 3.4 | 7.68 | 3.84 | 3595 | 128 | 2100 | 2500 |
| SBL-300-7D | 300 | 7D | 0.05 | −0.004 | +380 | 0 | 10.07 | 15.4 | 3.84 | 3.84 | 3624 | 96 | 700 | 2500 |

In the two NBL-cases **NBL-700-5D** and **NBL-700-7D** the turbine spacing is set to $s = 5D$ and $s = 7D$, where $D$ is the rotor
diameter of the turbine. The turbine spacing for all other cases is $s = 7D$. The NBL is capped by an inversion layer with a lapse
rate of $\Gamma = +1 \, \text{K km}^{-1}$ to achieve a BL height of approximately 700 m, which is a very common BL height in the German
Bight, according to the COSMO-REA6 climatology (cf. Fig. A3 and A4). The correct term for such an inversion-capped NBL
is *conventionally neutral boundary layer* (CNBL). However, the cases are named NBL-700-7D and NBL-700-5D to avoid
confusion with the CBL-cases.

Because CBLs are more frequent and are generally thicker than SBLs in the German Bight, two CBL-cases **CBL-700-7D**
and **CBL-1400-7D** with a BL height of $h \approx 700$ m and $h \approx 1400$ m, respectively, are simulated. This represents the spread of
CBL heights in the German Bight (cf. Fig. A3). Note that a CBL is the only BL type for which the BL height can be controlled





freely by the initial temperature profile without the need to change other parameters. The (steady-state) BL height of CNBLs and SBLs can not be controlled directly but is rather a function of friction velocity, Coriolis parameter, free atmosphere (FA)

stratification and surface buoyancy flux (Zilitinkevich et al., 2007).

The BL height of the SBL-case **SBL-300-7D** is $h \approx 300$ m, so that the wind turbines with a rotor top height of 270 m are still within the BL and do not penetrate into the FA. 300 m is a small but still typical value for an SBL in the German Bight (cf. Fig. A4).

The wind speed at hub height is set to 10 ms$^{-1}$ for all cases. This wind speed is less than the mean wind speed in the German

Bight (10.8 ms$^{-1}$, cf. Fig. A1) to stay below the rated wind speed of $v_{rated} = 10.59$ ms$^{-1}$ of the IEA 15 MW reference wind turbine (Gaertner et al., 2020). Thus the turbine operates at a high thrust coefficient and the turbine power is a function of the wind speed. The surface roughness length in all cases is $z_0 = 1$ mm. The wind direction is set to 225°, which is one of the most common wind directions in the German Bight. Because the main axis of the wind farm cluster in Zone 3 has a southwest-northeast orientation, strong wake effects can be expected for this wind direction.

## 2.3 Setup, boundary conditions, domain and wind farm layout

The domain and wind farm layout are shown in Fig. 2. The domain length and width are $L_x = 204.8$ km and $L_y = 163.84$ km, respectively. These lengths correspond to $n_x = 10240$ and $n_y = 8192$ grid points in x- and y-direction for isotropic grid spacings of $\Delta x = \Delta y = \Delta z = 20$ m for all cases. These spacings yield a density of 12 grid points per rotor diameter, which is enough to resolve the most relevant eddies inside the wind turbine wakes. Above the BL, where no turbulence must be re-

solved, the grid is stretched vertically to a maximum of $\Delta z_{max} = 50$ m to save computational cost. The stretch factor is $f_{stretch} = \Delta z(k+1)/\Delta z(k) = 1.08$ and the stretching starts at $z_s$ (cf. Table 1). To damp gravity waves before they could be reflected at the domain top, Rayleigh damping is applied above the Rayleigh damping level $z_{rd}$ with a Rayleigh damping factor of $f_{rd} = 0.01/\Delta t$, where $\Delta t$ is the time step. The domain height $L_z$, number of vertical grid points $n_z$, stretch level and Rayleigh damping level are different for the 5 cases and are given in Table 1. The simulated time in all 5 cases is 10 h. The first

6 h are required to obtain a steady-state wind farm flow (6 h is approximately the time that the flow needs to pass the domain, i.e. 204.8 km/10 ms$^{-1} \approx 5.7$ h). The last 4 h are used for the evaluation, e.g. averaging and flux calculations.

At the crosswise lateral boundaries, cyclic boundary conditions are applied and at the outflow plane, radiation boundary conditions are applied. At the inflow plane, steady-state vertical profiles of a precursor simulation are prescribed (details about the precursor simulations are given in the next section). To have a turbulent and stationary inflow from the beginning of the

main simulation, the flow field is initialized by the instantaneous flow field of the last time step of the precursor simulation. Because the precursor domain is much smaller than the main domain, the flow field is filled cyclically into the main domain. It is important to note that the width of the main domain is a non-integer multiple of the width of the precursor domain, to trigger the break-up of the unnatural periodicity in y-direction of the flow field, that is introduced by the cyclic fill method.

The turbulent state of the inflow is maintained by a turbulence recycling method, that maps the turbulent fluctuations from

the recycling plane at $x = x_r$ onto the inflow plane at $x = 0$ (Lund et al., 1998; Kataoka and Mizuno, 2002). The turbulent fluctuation $\Psi'(y,z,t)$ at each time step is defined as the difference between the absolute value $\Psi(x_r,y,z,t)$ and the horizontal

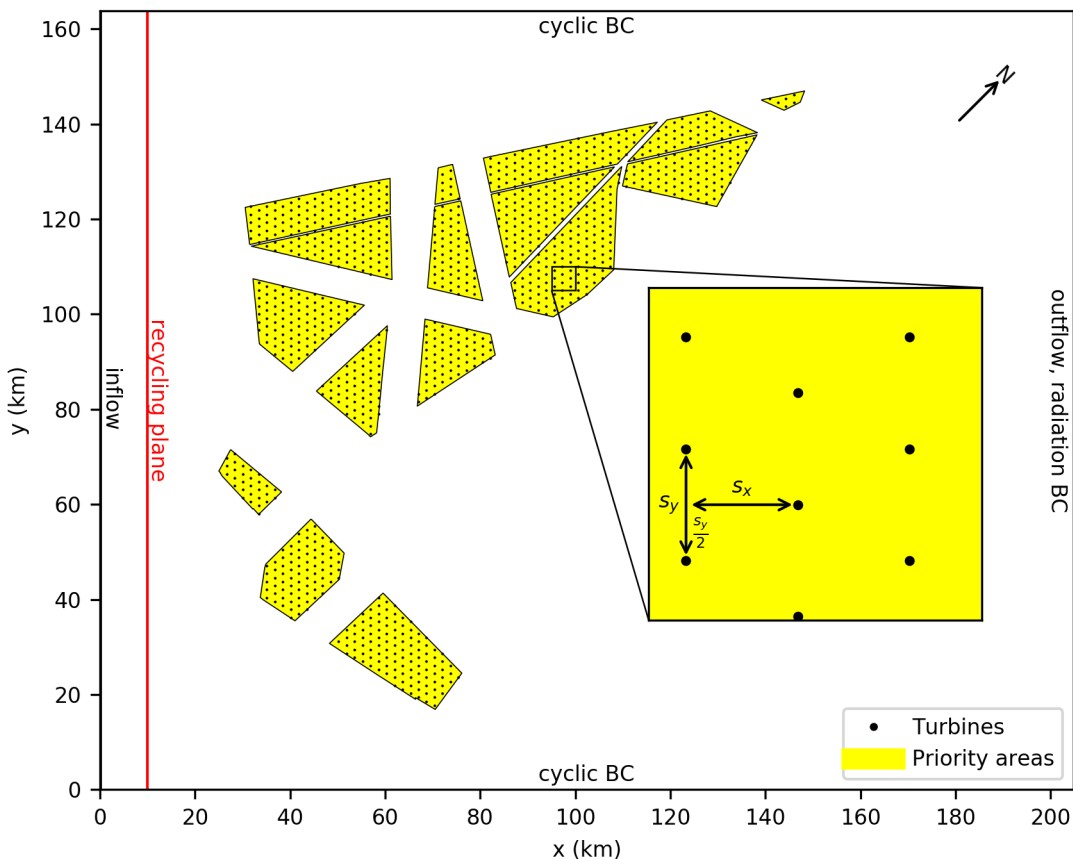

**Figure 2.** Domain and wind farm layout: Inflow from left, turbulence recycling plane at $x = 10$ km. Priority areas for future wind farms (cf. Fig. 1) are filled with a regular, staggered grid of wind turbines with a streamwise and crosswise turbine spacing of $s_x = s_y = 7$ D (shown here) or $5$ D.

line average in y-direction $\langle \Psi(x_r, z, t)\rangle_y$ at that height:

$$\Psi'(y, z, t) = \Psi(x_r, y, z, t) - \langle \Psi(x_r, z, t)\rangle_y, \tag{4}$$

where $\Psi$ can be a velocity component, the potential temperature or the subgrid-scale turbulent kinetic energy. The turbulent
fluctuation is added to the mean inflow profile $\Psi_{inflow}(z)$ at the inflow plane. Instead of adding it at the same y-location, it
can be added at $y + y_{shift}$:

$$\Psi(0, y + y_{shift}, z, t) = \Psi_{inflow}(z) + \Psi'(y, z, t). \tag{5}$$



The application of the y-shift effectively reduces the strength of streamwise elongated streaks in the mean wind speed of NBLs (Munters et al., 2016).[1] The otherwise inhomogeneous inflow with crosswise variations of wind speed of up to 10% would
hamper the evaluation of the wind farm power output and wake. A homogeneous inflow wind speed is of utmost importance in wind energy studies, because the wind turbine power is proportional to the third power of the wind speed. The y-shift is chosen in such a way, that the flow is recycled many times before reaching its initial y-position, which is achieved if the least common multiple of the y-shift and the domain width is a large number. The y-shift is also applied to the non-NBL cases, because it reduces crosswise variations of wind speed that are caused by wind farm induced flow blockage. The flow blockage leads to a
reduced mean wind speed at some y-locations of the recycling plane, which is "interpreted" as turbulent fluctuation, and thus mapped onto the inflow. Due to the self-reinforcing behaviour of this process, the crosswise variations can build up quickly without y-shift, even if the wind farms have a distance of 15 km to the recycling plane.

The turbulence recycling is limited to a height just above the BL height, so that potential BL growth between inflow and recycling plane will not affect the inflow BL height. The recycling plane is located 10 km downstream of the inflow plane,
which gives the turbulent structures enough time to interact and decorrelate before becoming recycled. For the CBL-cases, the absolute value of the potential temperature is recycled instead of its turbulent fluctuation, so that the inflow temperature rises according to the increasing surface temperature. This method is not needed in the SBL-case, because the surface temperature is constant in time (details in the next section).

The priority areas of Fig. 1 are rotated $45°$ clockwise, so that the inflow at hub height is parallel to the x-axis for a wind
direction of $225°$. The priority areas are filled with a regular array of the IEA 15 MW reference wind turbine, that has a rotor diameter of $D = 240$ m, a hub height of $z_{hub} = 150$ m and a rated power of $P_{rated} = 15$ MW (Gaertner et al., 2020). The wind turbines are staggered, i.e. every second column is shifted by half a turbine spacing in y-direction (cf. Fig. 2). The staggered configuration represents the real-world variation of wind directions better than the very special case of an aligned configuration. Additionally, power output and wake strength are less sensitive to potential wind direction changes (that might
occur further downstream inside the wind farm) for the staggered configuration, as revealed by own test simulations with smaller wind farms. The turbine spacing in the x- and y-direction is the same ($s_x = s_y = s$). The total number of wind turbines is $n_{wt} = 1063$ for $s = 7D$ and $n_{wt} = 2088$ for $s = 5D$, resulting in a total installed wind farm capacity of 15.9 GW and 31.3 GW, respectively. With a total wind farm area of $3000\,\mathrm{km}^2$, the resulting installed power density is $P''_{7D} = 5.3\,\mathrm{MW\,km}^{-2}$ and $P''_{5D} = 10.4\,\mathrm{MW\,km}^{-2}$. Note that $s = 7D$ and $P'' = 5\,\mathrm{MW\,km}^{-2}$ are typical values for currently existing wind farms
in the German Bight but that even with $s = 5D$ the total installed wind farm capacity stays below the 2040-expansion target of 40 GW. Note also that, for the sake of simplicity, all existing wind turbines in the priority areas are replaced by the much larger 15 MW wind turbine.

---

[1]Elongated, streak-like structures in the instantaneous streamwise wind speed (also called superstructures or Very-Large-Scale Motions) are a natural phenomenon of NBLs. However, these structures can be as large as 20 times the BL height (Fang and Porté-Agel, 2015), so that they can not be captured between inflow and recycling plane. Thus, the same structure is recycled repeatedly without breaking up or moving in y-direction. As a result, streaks of high and low wind speed appear in the averaged velocity field even for very long averaging times. A y-shift does not avoid the appearance of streaks in the instantaneous velocity field but due to the changing y-location of the streaks the strength of the streaks in the mean velocity field is reduced effectively (Munters et al., 2016).





## 2.4 Precursor simulations

Steady-state inflow profiles and a turbulent flow field for each main simulation are obtained by a precursor simulation with
cyclic boundary conditions in both lateral directions. In order to save computational time, the precursor domains are much
smaller than the main domain (cf. Table 1). The domain sizes are different for the different cases in order to ensure that the
largest structures of each BL type are covered several times. The number of vertical grid points, the stretching and Rayleigh
damping levels are the same as in the corresponding main simulation. It is important that the turbulence *and* the mean flow
are stationary at the end of the precursor simulation. If the mean flow that is prescribed at the inflow plane is not in steady
state it will try to reach it during its passage through the main domain, causing streamwise changes in mean quantities such
as wind speed and direction. While steady-state turbulence is reached after only a few hours, achieving a steady-state mean
flow can take several days due to the slow decay of the inertial oscillation, which has a period of 14.6 h at a latitude of $55°$ N.
The physical simulation times of the precursor simulations are 96 h for the cases NBL-700-7D, NBL-700-5D and CBL-1400-
7D, 48 h for the case CBL-700-7D and 24 h for the case SBL-300-7D. The initial velocity and potential temperature field is
horizontally homogeneous. Horizontal velocity components $u$ and $v$ are set to the geostrophic wind components $u_g$ and $v_g$ at
all heights. The geostrophic wind is adjusted so that the final wind speed at hub height is $10.0 \, \mathrm{m \, s^{-1}}$ and the wind direction
at hub height is parallel to the x-axis (cf. Fig 3). The onset of turbulence is triggered by small random perturbations in the
horizontal velocity field below a height of 150 m for the case SBL-300-7D and below 250 m for all other cases. The subgrid-
scale model of Dai et al. (2021) is used for the case SBL-300-7D. A grid convergence study showed that a grid spacing of 20 m
is sufficient, if this SGS-model is used, whereas the results are very grid spacing sensitive if the standard-SGS model of PALM
is used. Further setup details vary significantly between the different cases and hence are described separately in the following
sections.

### 2.4.1 NBL

The initial potential temperature profile of the NBL-cases is linear and has a vertical temperature gradient (lapse rate) of $\Gamma = +1 \, \mathrm{K \, km^{-1}}$ from the surface to the domain top (cf. Fig. 3). At the surface, a Neumann-condition for the potential temperature
is applied and the surface heat flux is set to zero. Shear-driven turbulence production leads to the formation of a neutrally
stratified BL that grows until it reaches a steady BL height of 780 m. The BL height is defined as the height at which the
shear stress reaches 5 % of its surface value. The conventionally neutral boundary layer is separated from the FA by a capping
inversion that has a stronger stratification than the FA.

### 2.4.2 CBL

The initial temperature profile of the CBL-cases consists of a constant potential temperature between the surface and the de-
sired BL height $h = 700$ m or $h = 1400$ m for the cases CBL-700-7D and CBL-1400-7D, respectively. Above that height the
potential temperature has a constant lapse rate of $\Gamma = +3.5 \, \mathrm{K \, km^{-1}}$, which corresponds to the International Standard Atmo-
sphere. A Dirichlet-condition is applied for the surface temperature and a constant surface heating rate of $\dot{\theta}_0 = +0.050 \, \mathrm{K \, h^{-1}}$

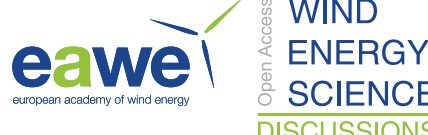

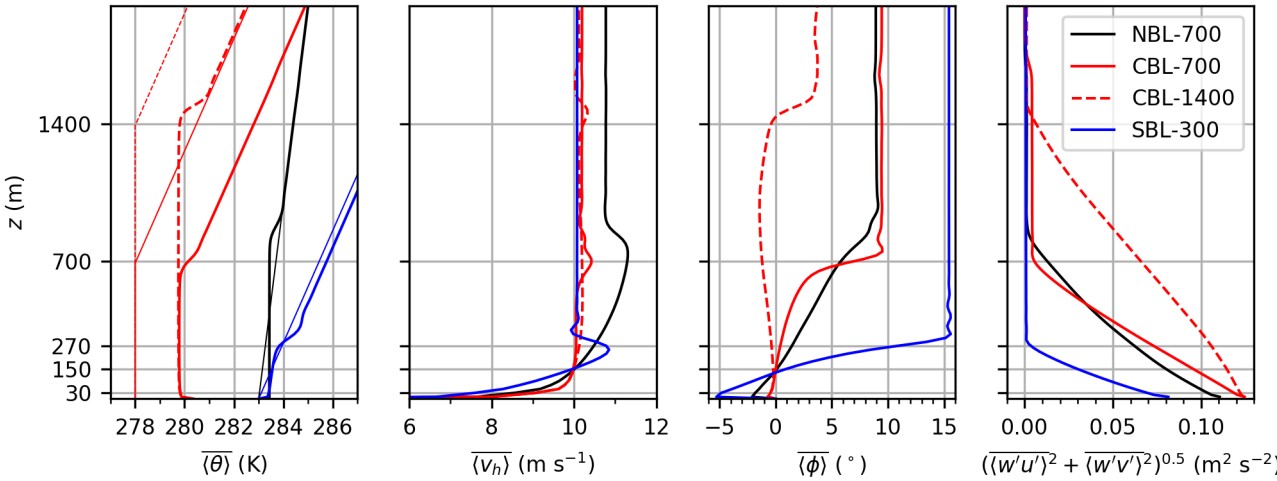

**Figure 3.** Vertical profiles of potential temperature ($\theta$), horizontal wind speed ($v_h$), wind direction ($\phi$, clockwise positive) and total (resolved + subgrid-scale) kinematic vertical momentum flux. The thin lines are initial profiles. The thick lines represent quantities that are horizontally averaged ($\langle \bullet \rangle$) over the entire precursor domain and temporal averaged ($\overline{\bullet}$) over the last hour of the precursor simulation. $\overline{\langle \theta \rangle}$, $\overline{\langle v_h \rangle}$ and $\overline{\langle \phi \rangle}$ are used as inflow profiles for the main simulations. BL heights 700 m and 1400 m as well as rotor top ($z = 270$ m), rotor bottom ($z = 30$ m) and hub height ($z = 150$ m) are marked on the vertical axis and with horizontal grey lines.

and $\dot{\theta}_0 = +0.025 \, \mathrm{K\,h^{-1}}$ is used to drive the CBL of the cases CBL-700-7D and CBL-1400-7D, respectively. The heating rates differ by a factor of 2 to achieve approximately the same surface heat flux $Q_0$ and Monin-Obukhov length $L$ (cf. Table 1), so that only the effect of a changing BL height is seen in the results.

Boundary layer growth is avoided by applying a large-scale subsidence that acts only on the potential temperature field. The subsidence velocity is zero at the surface and increases linearly to its maximum value $w_{sub}$ at the height $h$ and is constant

above. The subsidence velocity is chosen in such a way that the temperature increase in the FA exactly matches the surface heating rate. Thus the BL height can be kept precisely constant even for very long precursor simulations. Final BL heights, according to the definition given in section 2.4.1, are 690 m and 1400 m.

Large-eddy simulations of CBLs are usually driven by a constant heat flux, i.e. a Neumann-condition for the surface temperature. However, we decided to use a Dirichlet-condition, because of two reasons:

– It allows for spatial variations in the surface heat flux which may be caused by enhanced mixing inside the wind farms. In reality, the resulting change in sea surface temperature (on the scale of hours) would be very small due to the good turbulent mixing inside the ocean mixed layer during strong winds and due to the high heat capacity of water in contrast to that of air. Thus, it is more realistic to prescribe a horizontal homogeneous surface temperature than a horizontal homogeneous heat flux.





– Driving the CBL with a constant surface heating rate has the advantage that the temperature evolution inside the BL is
      known in advance and thus the subsidence velocity required for obtaining a constant BL height is also known in advance
      and does not have to be found iteratively.

### 2.4.3  SBL

The initial potential temperature profile of the SBL-case is linear and has a vertical temperature gradient (lapse rate) of $\Gamma =$
$+3.5 \, \mathrm{K \, km^{-1}}$ from the surface up to the domain top. A Dirichlet-condition is applied for the surface temperature, as this is
the correct surface forcing method for SBLs (Basu et al., 2008). Generating a steady-state SBL is not as simple as it is for the
CBL. A straight-forward method would be to use a surface cooling rate. However, due to the long simulation time, required for
the decay of the inertial oscillation, the elevated inversion at the top of the SBL would become unrealistically strong (Kosović
and Curry, 2000). We developed a method to generate a steady-state SBL in which the potential temperature profile is constant
in time and the strength of the elevated inversion can be freely adjusted.

The method uses the large-scale forcing functionality of PALM. Instead of changing the surface temperature, a positive
temperature tendency of $+0.05 \, \mathrm{Kh^{-1}}$ is added at every grid point and at every time step. This added tendency imitates a large
scale advection of warm air and thus forms an SBL with steady heat flux and momentum flux profiles. The heat flux divergence
results in a cooling tendency that exactly balances the positive large-scale advection tendency, so that the temperature inside
the BL stays constant. In the overlying inversion, the heat flux divergence decreases approximately linearly until it reaches
zero at the transition to the FA. Consequently, the temperature in the FA increases further and the overlying inversion becomes
stronger. To prevent further strengthening of the overlying inversion, the large-scale advection tendency is set to zero in the FA
at $t = 6 \, \mathrm{h}$ . Inside the overlying inversion, the large-scale advection tendency increases linearly to its maximum value inside
the BL, so that it approximately compensates the cooling tendency caused by the heat flux divergence. From that point on
the potential temperature profile is steady and the simulation can run until the inertial oscillation has decayed. Because the
potential temperature in the FA changes over time, it is excluded from the Rayleigh damping. Despite of the shallow BL, a
large domain height of $L_z = 3624 \, \mathrm{m}$ is used to capture gravity waves that are triggered by the wind farms. The final BL height,
according to the definition given in section 2.4.1, is 270 m.

### 2.5  Data analysis

Statistical data that are presented in the results section are obtained in the last 4 h of the 10 h main simulations. Temporal
averages are denoted by an overbar (e.g. $\overline{v_h}$) and horizontal averages by angled brackets (e.g. $\langle \theta \rangle$). The temporal averaged
horizontal wind speed $\overline{v_h}$ is calculated as the average of the absolute values of the wind vector:

$$\overline{v_h} = \overline{\sqrt{u^2 + v^2}} . \tag{6}$$

Resolved turbulent fluxes of momentum are calculated with the temporal eddy-correlation (EC) method. The correlation of two
turbulent quantities (e.g. $u' = u - \overline{u}$ and $w' = w - \overline{w}$) can not be calculated directly during the simulation, because the respective



mean quantities are not known in advance. However, the resolved turbulent flux can be calculated after the simulation if the correlation of the absolute quantities is calculated during the simulation:

$$\overline{w'u'} = \overline{(w - \overline{w})(u - \overline{u})} = \overline{wu - \overline{w}u - \overline{u}w + \overline{w}\,\overline{u}} = \overline{wu} - \overline{w}\,\overline{u} - \overline{u}\,\overline{w} + \overline{w}\,\overline{u}$$

$$\overline{w'u'} = \overline{wu} - \overline{w}\,\overline{u}. \tag{7}$$





## 3 RESULTS

The presentation and discussion of the results is divided into the two sections wake properties and power output. In the first section, the wake properties of very large wind farms and their effect on the BL flow is discussed. In the second section it is discussed how the power output of very large and small wind farms is affected by the variation of the turbine spacing and the meteorological conditions.

### 3.1 Wake properties

#### 3.1.1 Wind speed and wind direction at hub height

The mean horizontal wind speed at hub height is shown in Fig. 4 for all cases. Streamlines indicate the wind direction. For the NBL-cases, the wind speed is reduced from $10 \mathrm{~m\,s^{-1}}$ to $7 \mathrm{~m\,s^{-1}}$ for a turbine spacing of $s = 7$ D and $5 \mathrm{~m\,s^{-1}}$ for $s = 5$ D inside the large wind farms in Zone 3. The wake length is defined as the distance between the wind farm trailing edge and the point at which the wind speed recovers to 90 % of its initial value, i.e. $9 \mathrm{~m\,s^{-1}}$. For the small wind farms N-1, N-2 and

N-3 (cf. Fig. 1), the wake length ranges from $1$ km to $20$ km. However, the wake length of the large wind farms in Zone 3 is approximately $100$ km for $s = 7$ D and the wake extends beyond the model domain for $s = 5$ D.

The wake flow is deflected counterclockwise. The largest deflection angle of approximately $10°$ is observed for the smaller turbine spacing ($s = 5$ D). The counterclockwise wake deflection is consistent with the findings of Allaerts and Meyers (2016), who observed a counterclockwise deflection of $2° - 3°$ for $15$ km long wind farm. A counterclockwise wind direction change

(higher ageostrophic wind component) has also been observed by Abkar and Porté-Agel (2014) and Johnstone and Coleman (2012), who investigated infinitely large wind farms. The wake deflection is caused by a reduced Coriolis force, as it is shown in the next section. Because the Coriolis force is proportional to the wind speed, the deflection angle is higher for the case with the greater speed deficit (NBL-700-5D). The reasons for the slow speed recovery and the wake deflection are discussed in detail in the next section.

The inflow wind speed has slight variations in the crosswise direction, which are caused by the wind farm induced flow deceleration reaching the recycling plane (see section 2.3). The variations have an amplitude of approximately $0.1 \mathrm{~m\,s^{-1}}$, which is 1 % of the inflow wind speed.

For the CBL-cases, the wind speed is reduced to $6.5 \mathrm{~m\,s^{-1}}$ for $h = 700$ m and $8 \mathrm{~m\,s^{-1}}$ for $h = 1400$ m inside the large wind farms in Zone 3. Also, the wake length of the large wind farms is much longer for the shallow BL than for the thick BL. This

BL height dependency occurs because a thicker BL contains more kinetic energy that can be transported down to the wind turbine level by turbulent vertical mixing than a shallow BL. The wake length and speed deficit of small wind farms (e.g. N-1 and N-2) is relatively unaffected by the BL height because the wind farm induced internal boundary layer does not reach the top of the BL.

The wake is deflected counterclockwise in the CBL-cases, as well. The deflection angle is approximately $5°$ for the case

with the shallow BL and $1 - 2°$ for the case with the thick BL. The higher deflection angle for the case with the shallow BL is caused by a greater speed deficit compared to the case with the thick BL.





**Figure 4.** Mean horizontal wind speed $\overline{v_h}$ at hub height for all five cases (a-e) and perturbation pressure $p_*$ at hub height relative to its value at the inflow for the case SBL-300-7D (f). Streamlines indicate the wind direction.





A comparison between the cases NBL-700-7D and CBL-700-7D shows that the speed deficit inside Zone 3 is greater for the CBL-case. Also, the wakes of the small wind farms are longer for the CBL-case. This is contradictory to the well known fact that wind turbine and wind farm wakes are generally shorter in CBLs than in NBLs and SBLs (Porté-Agel et al., 2020, sec. 2.3 and 3.4.2). To achieve the same hub height wind speed for both cases, the geostrophic wind speed is 6 % greater for the NBL-case (cf. Table 1) than for the case CBL-700-7D. Additionally, the wind speed is supergeostrophic in the upper half of the NBL and thus the mean BL wind speed is approximately 10 % greater in the NBL-case than in the CBL-case. Stability likely has little-to-no affect, because the stratification of the CBL-case is only weakly unstable ($L = -420$ m, cf. Table 1).

In the stable case SBL-300-7D, the wind speed is reduced to below 7 m s$^{-1}$ in the first 20 km of the large wind farms in Zone 3. The wind speed deficit is greater and the wake is more than 20 km longer for the small wind farms, compared to the other cases with $s = 7$ D. The wake of the large wind farms in Zone 3, however, is not longer than in the cases NBL-700-7D and CBL-700-7D. This occurs because the speed recovery in the wake of this large wind farms is not driven by momentum flux divergence (which is stability dependent) but rather by a favorable pressure gradient (details are given in the next section). The case SBL-300-7D covers several flow features that can not be seen in the other cases. These features are namely: flow blockage in front of the wind farms, flow deflection around the wind farms and flow acceleration beside the wind farms and/or wakes. These features are related to the pressure field inside and around the wind farms. The perturbation pressure $\overline{p^*}$, relative to its value at the inflow, is shown in Fig. 4f. A high pressure region in the upstream part of the large wind farms in Zone 3 leads to an adverse pressure gradient and thus flow deceleration in front of the wind farms. This effect is known as blockage effect or flow blockage (Wu and Porté-Agel, 2017). At a distance of 2.5 D upstream of the first wind turbine row of the wind farms in Zone 3 the wind speed is reduced by approximately 10 % relative to the inflow wind speed . Wu and Porté-Agel (2017) reported 11 % speed reduction 2.5 D upstream of the first turbine row of a 20 km long wind farm in a CNBL with a FA stratification of $\Gamma = +5$ K km$^{-1}$. However, for $\Gamma = +1$ K km$^{-1}$ they reported a speed reduction of only 1.2 %, because the flow is subcritical (Froude number $Fr < 1$). Using the same definition[2] as in Wu and Porté-Agel (2017), the Froude number in the case SBL-300-7D is $Fr = 1.73$. Hence, the flow is supercritical and a strong blockage effect occurs.

In the downstream part of the large wind farms, a favorable pressure gradient tends to accelerate the flow, countering the wind turbine induced flow deceleration. Consequently, the wind speed does not decrease further but remains nearly constant at approximately 6 m s$^{-1}$. In the wake, the pressure is more than 5 Pa smaller than the undisturbed pressure upstream of the wind farms. This results in a relatively fast speed recovery in the wake and to wind speeds well above the inflow wind speed beside the wakes. Note that this effect might be overestimated because the wind farms block a relatively large fraction of the domain width. The pressure perturbations are induced by large scale gravity waves that are triggered by the wind farms. The observed pressure distribution in the streamwise direction is consistent with the findings of Allaerts and Meyers (2017) and Wu and Porté-Agel (2017), who investigated semi-infinite wind farms in CNBLs. The effect can only be seen in the case SBL-300-7D, because it is most extreme if the BL height approaches the total height of the wind turbines. More details about wind farm induced gravity waves are provided in section 3.1.4.

---

[2]For details about the calculation of the Froude number refer to Wu and Porté-Agel (2017) and Vosper et al. (2009).





Because the wind farms in this study also have a finite size also in the crosswise direction, it can be seen that the pressure
perturbation also significantly affects the wind direction. Due to the streamwise reduction in wind speed, the flow diverges in
the crosswise direction inside the wind farms. In the wake, where the flow accelerates, horizontal convergence can be observed.

### 3.1.2    Reasons for wake deflection and slow speed recovery

What is the reason for the slow speed recovery and the wake deflection inside and behind the large wind farms? In order to an-
swer that question, Fig. 5 shows streamwise (parallel to streamlines) and crosswise (perpendicular to streamlines) components
of the pressure gradient force[3], the Coriolis force $F_c$ and the resolved vertical turbulent momentum flux divergence, also called
frictional force $F_f$, at $z = 150$ m and $y = 120$ km. The pressure gradient force can be divided into the geostrophic pressure
gradient force $F_{gp}$, which is constant and is defined by the geostrophic wind, and the perturbation pressure gradient force $F_{pp}$,
that can vary horizontally due to wind farm induced pressure perturbations. The forces are averaged over 4 turbine spacings
along $x$ and 2 turbine spacings along $y$ in order to eliminate peaks in $F_{pp}$ that are caused by single turbines. Thrust forces of
the turbines are not included. The analysis is made from a Lagrangian frame of reference, examining the forces on an air parcel
during its passage through the wind farms. From an Eulerian frame of reference, the sum of all forces, including the advection
tendencies, would sum to zero, because the flow is stationary.

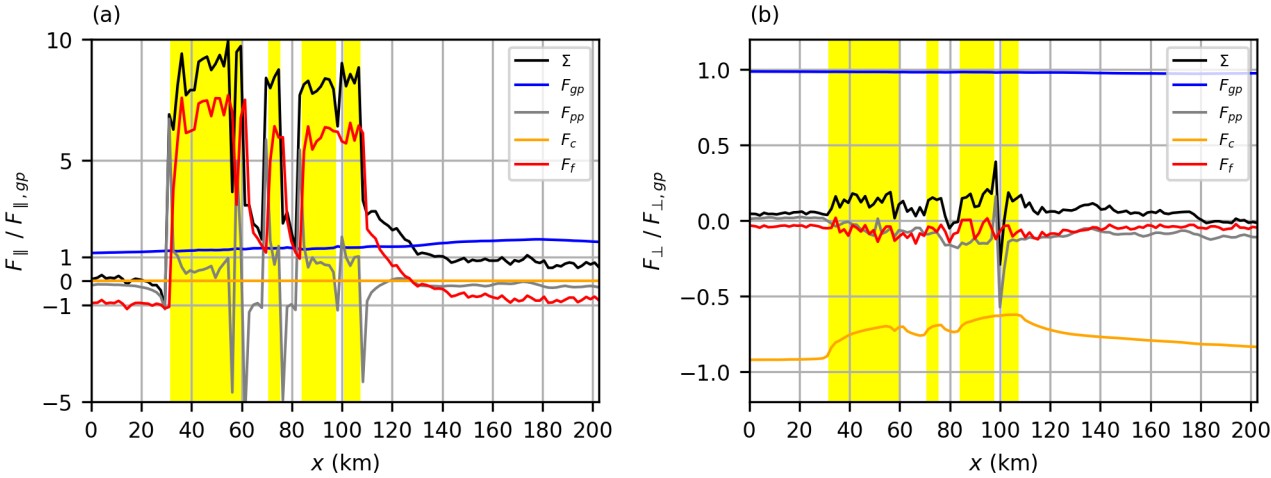

**Figure 5.** Streamwise and crosswise force components $F_\parallel$ (a) and $F_\perp$ (b) along a line at $y = 120$ km, $z = 150$ m for the case NBL-700-7D.
Shown are the geostrophic pressure gradient force ($F_{gp}$), the perturbation pressure gradient force ($F_{pp}$), the Coriolis force ($F_c$), the frictional
force ($F_f$, momentum flux divergence) and the sum of all forces ($\Sigma$). The forces are normalized by the respective geostrophic pressure
gradient force component at the inflow and are horizontally averaged over one turbine spacing along $x$ and $y$. The position of the wind farms
is marked by yellow areas.

---

[3]The pressure gradient force and the Coriolis force are not considered explicitly in the model, but are considered implicitly by the geostrophic wind.



Streamwise force components in Fig. 5a show that the accelerating geostrophic pressure gradient force and the decelerating
momentum flux divergence are in balance and sum to zero at the inflow. The streamwise component of the Coriolis force is
zero because this force acts always perpendicular to the flow. Inside the wind farms, the momentum flux divergence is positive
and thus is an accelerating component. It is the dominant driving force because it is more than seven times greater than the
geostrophic pressure gradient force.

An increasing perturbation pressure in front of the wind farm leads to a negative perturbation pressure gradient force and
thus flow deceleration (often called blockage effect). However, inside the wind farms the perturbation pressure gradient force
is positive due to a favorable pressure gradient (decreasing pressure). In the near wake the momentum flux divergence is high
and leads to a fast speed recovery. The momentum flux divergence decreases fast until it becomes negative in the far wake,
so that the speed recovers slowly in the far wake. The only force that remains for driving the flow is the geostrophic pressure
gradient force. At the inflow, this force is in balance with the momentum flux divergence, but in the wake this is not the case
due to two reasons: first, the negative momentum flux divergence is weaker than at the inflow due to a smaller wind speed
and thus a reduced near-surface momentum flux. Second, the streamwise component of the geostrophic pressure gradient
force has increased because the wake flow is deflected counterclockwise (i.e. to lower pressure). These results show that the
wake deflection is an elementary feature of the wake that supports the wind speed recovery. They also show that mixing of
momentum from the BL to the wind turbine level is not the dominant process that drives the speed recovery in the far wake of
very large wind farms.

The wake deflection can be explained by examining the crosswise force components that are shown in Figure 5b. Positive
forces result in counterclockwise flow deflection and negative forces result in clockwise flow deflection. At the inflow, the
Coriolis force, the geostrophic pressure gradient force and the momentum flux divergence are in balance. Because the Coriolis
force is proportional to the wind speed, it is reduced by approximately 30 % inside the wind farms and the wake. Consequently,
the sum of all forces becomes positive and the flow is deflected counterclockwise. The momentum flux divergence and the
perturbation pressure gradient force are negative inside the wind farm and inside the wake and are therefore opposing the wake
deflection. The negative perturbation pressure gradient force is a result of the pressure distribution around the wind farms that is
caused by the wind farm shape (cf. Fig.4d). The reason for the negative momentum flux divergence is the enhanced downward
mixing of negative y-momentum of the overlying flow, which is veered to the right (cf. Fig. 3). For small wind farms this
process can be dominant and may result in clockwise wake deflection (Van Der Laan and Nørmark Sørensen, 2017). However,
for very large wind farms, as in this study, the effect of the reduced Coriolis force is dominant. An appropriate parameter for
estimating the importance of Coriolis effects is the Rossby number. Coriolis effects become dominant for Rossby numbers
close to or below 1. For the large wind farms in Zone 3 with a length of $L_{wf} \approx 100$ km at mid-latitudes (Coriolis parameter
$f \approx 10^{-4}$) and a wind speed of $U \approx 10 \mathrm{\ m\,s^{-1}}$ the Rossby number becomes:

$$Ro = \frac{U}{L_{wf} f} \approx 1 \,, \tag{8}$$

indicating that Coriolis effects play an important role for flows in wind farms of this size.





### 3.1.3 Turbulence intensity at hub height

The turbulence intensity TI is defined as in Porté-Agel et al. (2013):

$$\text{TI} = \frac{\sqrt{\frac{2}{3}\text{TKE}}}{\overline{v_h}}, \tag{9}$$

where TKE is the resolved turbulence kinetic energy defined as:

$$\text{TKE} = \frac{1}{2}(\overline{u'^2} + \overline{v'^2} + \overline{w'^2}), \tag{10}$$

where $\overline{u'^2}$, $\overline{v'^2}$ and $\overline{w'^2}$ are the resolved-scale variances of $u$, $v$ and $w$, respectively.

The TI at hub height is shown in Fig. 6. Inside the wind farms, the TI reaches a fully developed state after approximately 4 rows and is constant farther downstream. A smaller turbine spacing leads to a greater TI inside the wind farms. For the case
NBL-700-7D, a TI of 10 % is reached inside the wind farms, but more than 14 % is reached in the case NBL-700-5D. In the CBL-cases, the TI inside the wind farms reaches 10 % in case CBL-700-7D and approximately 12 % in CBL-1400-7D. Although the ambient TI is only approximately 3 % for the case SBL-300-7D, the TI inside the wind farms reaches values similar to NBL-700-7D (approximately 10 %).

The wake in terms of TI is generally shorter than the wake in terms of wind speed (cf. Fig. 4). The shortest wakes occur in
the shallow SBL (SBL-300-7D) and the longest wakes occur in the thick CBL (CBL-1400-7D). This is the opposite behaviour than that of the wake in terms of wind speed. The wake length in terms of TI weakly depends on wind farm size with slightly longer wakes for larger wind farms. However, this effect is caused by the definition of the TI, in which the wind speed variances are normalized by the mean horizontal wind speed. The mean horizontal wind speed is smaller in the wake of the large wind farms than in the wake of the small wind farms, resulting in a higher TI in the wake of the large wind farms. The wind farm
size dependency of the wake length vanishes if the TKE is used for measuring the wake length instead of the TI (not shown).

In the NBL-cases and especially in the SBL-case, the TI in the far wake drops below the ambient TI at the inflow. This effect is caused by the reduced wind speed in the far wake which leads to a reduction in the shear-driven turbulence production. In the CBL-cases, there is also buoyancy-driven turbulence production, which is unaffected by the reduced wind speed in the wake and thus maintains the TI level. That buoyancy-driven turbulence production has a large impact on hub height TI is also
verified by the fact that the ambient TI is greater in the CBL-case with the thick BL (CBL-1400-7D) than in the case with the shallow BL (CBL-700-7D): The convective velocity scale $w^*$ is greater in the case CBL-1400-7D than in case CBL-700-7D and hence also the buoyancy-generated velocity variances are greater. Because buoyancy acts as a TKE-sink in the SBL-case, it can not compensate for the reduction in shear-driven turbulence production and thus the TI in the wake drops below 2 %. This effect is amplified by the entrainment of warm air into the BL, that leads to a stabilization (increased lapse rate) at hub
height and therefore stronger turbulence damping (cf. Fig. 6f and Fig. 8). The entrainment of warm air results in a temperature increase at hub height of approximately 0.3 K for the large wind farms and approximately 0.1 K for the small wind farms.



**WIND
ENERGY
SCIENCE
DISCUSSIONS**

**Figure 6.** Turbulence intensity (TI) at hub height for all five cases (a-e) and potential temperature at hub height relative to the inflow temperature for the case SBL-300-7D (f).



### 3.1.4 Boundary layer development

Figure 7 shows vertical cross sections of the horizontal mean wind speed for all cases. The cross sections are located at $y = 120$ km and thus cross the large wind farms in Zone 3. The inversion layer height $z_i$ is marked by lines at which the

maximum vertical potential temperature gradient occurs.

The inversion layer (IL) height is affected by the presence of the wind farms in all five cases. In the NBL-cases the IL is displaced upwards by $200 - 300$ m, whereas a larger displacement occurs for the smaller turbine spacing. Because the mean wind speed inside the BL decreases in the streamwise direction, the IL must be displaced upwards in order to maintain a constant mass flux inside the BL. The increase in IL height is not caused by entrainment of warm air into the BL (as can also

be seen in the profiles of potential temperature in Fig. 8). This phenomenon has also been observed by Allaerts and Meyers (2017), who also stated that the mass flux conservation is the reason for the IL displacement. Abkar and Porté-Agel (2014) stated that a smaller turbine spacing results in a larger BL height for an infinite wind farm in a CNBL. The IL displacement causes an acceleration of the flow in the FA for the NBL and CBL cases- Details about this effect are described in the next section.

In the CBL-cases, the IL height increases above the wind farms and decrease above the wake, reaching its initial value at approximately 70 km downstream of the last wind farm trailing edge. The IL displacement is larger for the shallower BL, i.e. the case CBL-700-7D.

The IL displacement is most significant for the case SBL-300-7D. The IL height increases from $300$ m to $500$ m. Allaerts and Meyers (2016) also reported that larger IL displacements occur for shallower BLs ($+60$ % for $h = 250$ m). In the case

SBL-300-7D, the IL height increase is caused by vertical displacement due to mass conservation and also by entrainment of warm air into the BL (see Fig. 7f). The entrainment of warm air into the BL leads to a warming of the lower part of the BL. However, the temperature at the height of the original IL is reduced because the warm air in the IL is replaced by relatively cold air from the BL.

Because the laminar flow in the FA is adiabatic, the isotherms in Fig. 7f can be interpreted as streamlines. They show that

gravity waves are excited by the wind farms. There are small scale gravity waves with a wave length that corresponds to the turbine spacing and a large scale gravity wave with a wave length that approximately corresponds to the wind farm length. The negative and positive temperature deviations in the wave crest and trough, respectively, cause a positive and negative deviation in the perturbation pressure at the surface, as it is shown in Fig. 4f. A detailed analysis of the wind farm induced gravity waves goes beyond the scope of this study. However, it is noted that the qualitative pressure and temperature distributions correspond

to the findings of Allaerts and Meyers (2017) and Wu and Porté-Agel (2017).

Additional test simulations have shown that the strength of the gravity waves is sensitive to the domain height. Allaerts and Meyers (2017) achieved good results (low wave reflection at the domain top) if the domain height corresponds to at least one vertical wave length $\lambda_z = 2\pi U/N$, where $U$ is the BL bulk wind speed and $N$ is the Brunt-Vaisala frequency in the FA. In the case SBL-300-7D, the domain height is set to $0.43\lambda_z$ ($\lambda_z = 5.9$ km, $L_z = 3624$ m), because larger domain heights lead to

numerical instabilities at the inflow. Wu and Porté-Agel (2017) used a domain height of $L_z = 2.4$ km for a FA-stratifications

**Figure 7.** Vertical cross sections at $y = 120$ km of wind speed $v_h$ for all cases (a-e) and potential temperature deviation relative to the inflow temperature for case SBL-300-7D (f). The inversion layer height $z_i$ is marked by a line that marks the maximum vertical temperature gradient.





of $\Gamma = 1 \, \mathrm{K\,km^{-1}}$ (resulting in $L_z = 0.22\lambda_z$) and $\Gamma = 5 \, \mathrm{K\,km^{-1}}$ (resulting in $L_z = 0.49\lambda_z$). It is not clear whether the vertical wave length is the only relevant parameter for choosing the correct domain height or whether the wind farm length also has to be considered. Further research is needed to find setup guidelines that ensure that wind farm induced gravity waves are covered as realistically as possible.

### 3.1.5 Profiles of wind speed, wind direction and potential temperature in the wake

To examine the effect of the wind farms on the BL in more detail, profiles of wind speed, wind direction and potential temperature are shown in Fig. 8. The profiles are evaluated at the inflow ($x = 0$ km), in the near wake ($x = 120$ km) and in the far wake ($x = 180$ km) of the large wind farms in Zone 3 ($y = 120$ km).

The wind speed profiles show that the wind farm induced wind speed deficit spreads over the entire height of the BL. The effective vertical mixing in the CBL-cases results in an approximately height-constant wind speed at the inflow and in the wake. In the case CBL-1400-7D, the wind speed in the upper part of the BL is even lower than in the lower part of the BL at $x = 120$ km. In the NBL-cases, the vertical mixing is not as effective and thus a significant wind shear exists over the entire BL in the wake. The wind speed profiles of the case SBL-300-7D show that the BL has grown from $300$ m to $500$ m and that the super-geostrophic maximum is eliminated completely. The IL displacement causes an increase in wind speed in the FA above the BL. The maximum increases (approximately $1 \, \mathrm{m\,s^{-1}}$) are observed for the cases with the greatest IL displacements. That suggests that the wind speed excess above the BL is also caused by the continuity constraint, i.e. the wind speed has to increase in order to maintain a constant mass flux between the IL and the domain top. For the CBL-cases, where the IL height decreases again behind the wind farms, the wind speed above the BL decreases to below-geostrophic in the far wake ($x = 180$ km). Note that these effects could be overestimated, because of the artificial boundary that is introduced by the Rayleigh damping layer that starts several hundred meters above the BL. The sensitivity of this effect on the Rayleigh damping height has not been investigated because the scope of this study is on BL-internal effects.

The wake deflection shown in the horizontal cross sections can also be seen in the wind direction profiles. The wind farm induced wind direction change is approximately constant over the entire height of the BL. The largest deflection angles of up to $-10°$ are observed for the cases with the greatest speed deficit (SBL-300-7D and NBL-700-5D) because the Coriolis force reduction is greatest in these cases. The smallest deflection angle is observed in the case CBL-1400-7D with no deflection in the upper half of the BL. All cases have in common that, in contrast to the counter clockwise deflection in the BL, the flow in the FA is deflected *clockwise*. As a result, the wind veer in the inversion layer increases to approximately $10°$. The flow deflection in the FA is also a Coriolis effect. Because the wind speed in the FA is supergeostrophic, the Coriolis force is greater than the geostrophic pressure gradient force and therefore the flow is deflected clockwise. The largest deflection angle of more than $10°$ is observed for the case NBL-700-5D. For this case the wind speed excess in the FA is greatest and stays positive until $x = 180$ km. Thus the Coriolis force is greatest and acts on the flow for a long time so that a high deflection angle is achieved. Note that this effect might also be overestimated due to the potentially overestimated wind speed excess. In the case SBL-300-7D, the combination of clockwise deflection in the FA and counterclockwise deflection in the BL leads results in a total wind veer of approximately $40°$ between the surface and the FA.



**Figure 8.** Vertical profiles of temporal averaged horizontal wind speed ($\overline{v_h}$), wind direction ($\overline{\phi}$, clockwise positive) and potential temperature ($\overline{\theta}$) relative to surface temperature $\overline{\theta_0}$ at the inflow ($x = 0$ km), in the near wake ($x = 120$ km) and in the far wake ($x = 180$ km) of the large wind farms in Zone 3 ($y = 120$ km) for all five cases.





The effect of the wind farms on the potential temperature profiles is largest for shallow BLs (SBL-300-7D) and negligible
small for thick BLs (CBL-1400-7D). The potential temperature profiles inside the well-mixed BLs of the NBL- and CBL-cases
are nearly unaffected by the wind farms. The greatest changes take place in the inversion layer, which is displaced upwards
in order to maintain a constant mass flux in the BL, as already described in the previous section. The profiles show that the
potential temperature inside the BL is unchanged and thus BL-warming due to entrainment of warm air from the FA is *not* the
reason for the increased IL height. On the contrary, the temperature at the height of the original IL decreases by approximately
$0.5$ K because it is replaced by colder air from the underlying BL. The potential temperature profile of the SBL-case is heavily
modified by the wind farms. The temperature in the BL increases by approximately $0.5$ K due to entrainment of warm air from
the FA into the BL. The IL rises from $300$ m to $500$ m due to the combined effect of BL warming and IL displacement. Because
the surface temperature is constant, a new SBL forms in the far wake. This new SBL is shallower and more stably stratified
than the original SBL at the inflow.





## 3.2 Power output

### 3.2.1 Wind turbine and wind farm efficiencies

The effect of different turbine spacings, BL heights and stabilities on the power output of very large wind farms is investigated here. This is done by comparing wind farm efficiencies of small and large wind farms for the five simulated cases. Here, the
turbines in the area N-1 are defined as a small wind farm, because this area has the size of a typical, currently existing wind farm in the German Bight. The turbines in Zone 3 are defined as a large wind farm, because this areas will be equipped with wind farms in the future. (cf. Fig. 9f). The small wind farm consists of $27$ wind turbines for $s = 7$ D and $54$ wind turbines for $s = 5$ D, resulting in an installed wind farm capacity of $0.405$ GW for $s = 7$ D and $0.810$ GW for $s = 5$ D. The large wind farm consists of $636$ wind turbines for $s = 7$ D and $1260$ wind turbines for $s = 5$ D, resulting in an installed wind farm capacity
of $9.54$ GW for $s = 7$ D and $18.90$ GW for $s = 5$ D.

The wind farm efficiency $\eta_{wf}$ is defined as the total wind farm power $P_{wf}$ normalized by the wind farm power that would be achieved if all wind turbines $n_{wt}$ were operating in free-stream conditions, generating the reference power $P_{ref}$ (all quantities are averaged over the last $4$ h of the simulation):

$$\eta_{wf} = \frac{P_{wf}}{n_{wt}P_{ref}}. \tag{11}$$

The reference power is obtained by an additional simulation of a single turbine using the same inflow profiles as for the case NBL-700-7D. The reference power is $P_{ref} = 12.56$ MW. The wind farm efficiency can also be interpreted as the wind turbine efficiency averaged over all wind turbines of the wind farm. The wind turbine efficiency of a wind turbine generating $P_{wt}$ is defined as:

$$\eta_{wt} = \frac{P_{wt}}{P_{ref}}. \tag{12}$$

The wind farm efficiencies of the small and the large wind farm are listed in Table 2 and the wind turbine efficiencies are shown in Fig. 9.

**Table 2.** Wind farm efficiencies for a small wind farm (N-1) and a large wind farm (Zone 3) for all five cases.

| Case | Wind farm efficiency | |
| --- | --- | --- |
| | N-1 | Zone 3 |
| NBL-700-7D | 0.87 | 0.58 |
| NBL-700-5D | 0.77 | 0.41 |
| CBL-700-7D | 0.86 | 0.54 |
| CBL-1400-7D | 0.88 | 0.63 |
| SBL-300-7D | 0.61 | 0.42 |





**Figure 9.** Wind turbine efficiencies $\eta_{wt}$ for all five cases (a-e) and overview of wind farm names (f).



In general, the wind farm efficiency is significantly lower for large wind farms than for small wind farms. All 7D-cases, except for the SBL-case, show efficiencies of $0.86 - 0.88$ for the small wind farm and efficiencies of $0.54 - 0.63$ for the large wind farm. In the SBL-case, the efficiency of the small wind farm is $0.61$, because the wind farm is affected by the blockage effect of the sum of all wind farms. This is visible in Fig 9e, which shows that the efficiency of the wind turbines in the first row of N-1 is already below $0.8$. The efficiency of the large wind farm is $31\%$ lower than that of the small wind farm for the SBL-case. The blockage effect redistributes energy from upstream parts of the wind farm to downstream parts of the wind farm by a favorable pressure gradient, which has already been shown by Allaerts and Meyers (2017) for wind farms in shallow CNBLs. This effect can also be seen in the power distribution inside the farm: The turbine power is constant from approximately row 10 up to the trailing edge of the large wind farm in Zone 3 (cf. Fig. 9e and Fig. 10e). In all other cases the wind turbine power does not reach a steady state until the end of the wind farms.

A reduction of turbine spacing from $s = 7D$ to $s = 5D$ results in an efficiency reduction of $12\%$ ($0.87$ to $0.77$) for the small wind farm, but results in an efficiency reduction of $29\%$ ($0.58$ to $0.41$) for the large wind farm. The low wind farm efficiency for the case NBL-700-5D can be explained by a fast drop of the turbine efficiencies to values below $0.4$ only $20\,\mathrm{km}$ downstream of the leading edge. The low wind turbine efficiencies are caused by a reduction in the vertical kinetic energy flux, as shown in the next section.

A doubling of the BL height results in an efficiency increase of $+2\%$ (from $0.86$ to $0.88$) for the small wind farm but in an efficiency increase of $17\%$ (from $0.54$ to $0.63$) for the large wind farm. The dependency of wind farm efficiency on the BL height has also been observed by Allaerts and Meyers (2016), who reported a $17.6\%$ increase in power deficit for a BL height reduction from $1000\,\mathrm{m}$ to $250\,\mathrm{m}$.

A comparison between the cases NBL-700-7D and CBL-700-7D shows that greater wind farm efficiencies are obtained for the NBL, although better efficiencies are expected for the CBL due to the better vertical mixing. The wind farm efficiencies for the NBL-case are greater because the inflow wind speed in the bulk of the BL is higher for the NBL-case than for the CBL-case (cf. Fig. 3). This result shows that it is important to consider not only the wind speed at hub height, but also the wind profile inside the entire BL to make accurate wind farm performance predictions.

### 3.2.2 Energy flux analysis

To examine the dependency of the wind farm efficiency on the turbine spacing and the BL height in more detail, an energy flux analysis is made in this section. The analysis is a simplified version of the analyses made by Abkar and Porté-Agel (2014) and Allaerts and Meyers (2017). The extraction of kinetic energy by the wind turbines can be compensated for by two sources of energy:

1. Vertical turbulent flux of kinetic energy at rotor top level, $W_f$

2. Work done by the geostrophic pressure gradient on the flow below rotor top level (bottom of the BL), $W_{p,b}$



The resolved downward turbulent flux of mean kinetic energy at rotor top level, averaged between $y = 120 \text{ km} - s_y$ and $y = 120 \text{ km} + s_y$, is calculated by multiplying the shear stress with the corresponding wind velocity component at that height:

$$W_f(x) = \langle -\rho(\overline{u}\,\overline{w'u'} + \overline{v}\,\overline{w'v'})|_{z=z_t} \rangle_y .$$
(13)

and the power density of the energy input by the geostrophic pressure gradient on the flow below rotor top level $z_t = 270$ m is calculated as:

$$W_{gpg,wt}(x) = \int_{z=0}^{z_t} \rho f_c(u_g\overline{v}(z) - v_g\overline{u}(z))dz|_{y=120 \text{ km}} .$$
(14)

The work done by the geostrophic pressure gradient on the rest of the BL (between $z_t$ and $z_i$) is calculated as:

$$W_{gpg,BL}(x) = \int_{z=z_t}^{z_i} \rho f_c(u_g\overline{v}(z) - v_g\overline{u}(z))dz|_{y=120 \text{ km}} .$$
(15)

In Fig. 10 these power densities are compared with the power densities of the wind turbines:

$$W_{wt} = \frac{P_{wt}}{s_x s_y} .$$
(16)

For the case NBL-700-7D it can be seen that the first-row wind turbines operate at the reference power, so that a high power density of $W_{wt} = 12.56 \text{ MW}/(7D)^2 = 4.45 \text{ W m}^{-2}$ is achieved. The power density of downstream wind turbines is deter-
mined by the kinetic energy flux. Because the kinetic energy flux decays from 3 W m$^{-2}$ at the beginning of the first wind farm to 2 W m$^{-2}$ at the end of the last wind farm, the power density of the wind turbines also decays to below 2 W m$^{-2}$. The good correlation between the wind turbine power density and the kinetic energy flux has also been found by Stevens et al. (2016) for the fully developed regime in a 9 km long wind farm. The work done by the pressure gradient on the flow below the rotor top level achieves a power density of approximately 0.6 W m$^{-2}$. It is thus not the dominating energy source inside the wind farms
but it still contributes approximately 20 % to the total energy input $W_{vkef} + W_{gpg,wt}$. The wind turbines extract approximately 70 % of the total energy input $W_{kef} + W_{gpg,wt}$, which is a relatively large value. Johnstone and Coleman (2012) and Abkar and Porté-Agel (2014), who analyzed the energy budgets for an infinite wind farm in a NBL, reported that 35 % and 45 %, respectively, of the energy input by the pressure gradient is extracted by the wind turbines. The differences could be explained by the low Reynolds number of $Re = 1000$ in the simulations of Johnstone and Coleman (2012) and the higher roughness
length of $z_0 = 0.1$ m in the simulations of Abkar and Porté-Agel (2014).

Although the kinetic energy flux does not reach a constant value until the end of the wind farms it is likely that it approaches the power density of the work done by the pressure gradient on the BL flow above the wind turbine level $W_{gpg,BL}$. Therefore, the flow approaches the fully-developed regime of an infinite wind farm flow, where all the energy extracted by the wind turbines is provided by the work done by the pressure gradient on the BL flow (Johnstone and Coleman, 2012; Abkar and
Porté-Agel, 2014). The energy input by the pressure gradient ($W_{gpg,wt} + W_{gpg,BL}$) achieves power densities of only $1 - 2$ W m$^{-2}$, which is much smaller than the power density achieved by the first-row wind turbines. As the case NBL-700-5D

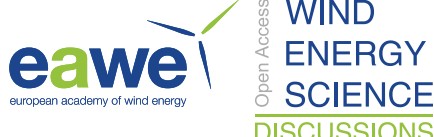

**Figure 10.** Comparison of power density provided by the pressure gradient below rotor top level $W_{gpg,wt}$ and above rotor top level $W_{gpg,BL}$, the vertical kinetic energy flux at rotor top level $W_{vkef}$ and the power density of the wind turbines located between $y = 120 - s_y$ and $y = 120 + s_y$. Panel (f) shows the ageostrophic wind speed component ($\overline{v_{h,a}}$) at hub height for the case NBL-700-7D at $y = 120$ km.





shows (Fig. 10b), a reduction of the turbine spacing from $s = 7$ D to $s = 5$ D results in a doubling of the power density of the first-row wind turbines, but the power density of the last row wind turbines is as low as for $s = 7$ D. This result indicates that the turbine spacing for very large wind farms should be chosen much larger than for small wind farms to achieve a good wind farm efficiency. That the wind farm power output is limited by the vertical kinetic energy flux has also been found by Badger et al. (2020), who investigated potential wind farm scenarios in the German Bight using a mesoscale weather forecast model (WRF) and a simple box model (KEBA). Nishino (2013) used a simple, theoretical approach to show that the power density of very large wind farms is limited by the energy input of the pressure gradient and that the power density is proportional to $\tau_0 U_h$, where $\tau_0$ is the shear stress near the surface and $U_h$ is the mean wind speed at hub height for an undisturbed flow without wind farms. However, Nishino (2013) neglects the effect of the wind farm on the flow inside the BL. According to Abkar and Porté-Agel (2014) and Eq. (15), the energy input by the pressure gradient depends on the BL height and on the ageostrophic wind speed component averaged over the BL. The BL height increases due to the presence of the wind farms and the ageostrophic wind speed component increases due to the counterclockwise wake deflection (cf. Fig. 10f). Consequently, the wind farm induced flow effects result in a significant increase of the energy input by the pressure gradient, as it can be seen in Fig.10a and b. This effect occurs only in the wake although the BL height and wind direction already change inside the wind farms. The reason is the decrease in the absolute wind speed that tends to reduce the ageostrophic wind speed component and thus compensates the increasing ratio of ageostrophic to geostrophic wind speed (counterclockwise wind direction change). In the wake, the wind speed recovers and thus the ageostrophic wind speed component becomes larger than upstream of the wind farms. The described effect is largest for case with the small turbine spacing NBL-700-5D, because the BL growth and the wake deflection angle is largest for this case.

Figures 10c and d show that a doubling of the BL height has approximately no effect on the energy input by the pressure gradient on the undisturbed inflow. The effect of the thicker BL is compensated by a smaller ageostrophic wind speed component inside the BL. This is indicated by a much smaller angle between hub height wind and the geostrophic wind of $\alpha = 3.4\,°$ for the case CBL-1400-7D than $\alpha = 9.5\,°$ for the case CBL-700-7D (cf. Table 1 and Fig. 3). Consequently, the ageostrophic wind speed component inside the BL of the stationary inflow adjusts in such a way that the resulting energy input by the pressure gradient balances the energy extraction by TKE production near the surface. As stated earlier, the power output of infinitely large wind farms is determined by the energy input of the pressure gradient. Hence, the power output of infinitely large wind farms does not depend on the BL height, at least for this idealized setups with a stationary CBL inflow. However, for very large, but finite-size wind farms, as in this study, a significant amount of the energy extracted by the wind turbines comes from the horizontal influx of mean kinetic energy inside the BL. As this influx is proportional to the BL height, much more energy is available for the wind turbines in the case CBL-1400-7D than in the case CBL-700-7D. This results in a higher kinetic energy flux and also a slower decay of the vertical kinetic energy flux for the case CBL-1400-7D than for the case CBL-700-7D, as shown in Fig. 10c and d. The same holds for the wind turbine power densities, because they are directly related to the vertical kinetic energy flux. Therefore, higher wind farm efficiencies are achieved for the thicker BL.

The case SBL-300-7D is very special, because the rotor top level matches the BL height of the inflow. Thus, the energy input by the pressure gradient above the rotor top level, as well as the vertical kinetic energy flux at the rotor top level, is zero





upstream of the wind farms. Both components become non-zero inside the wind farms due to the vertical displacement of the inversion layer (BL growth). However, the energy input by the pressure gradient below rotor top level remains the dominant energy source with approximately $1 \, \mathrm{W \, m^{-2}}$, except for the first $10 \, \mathrm{km}$ of the wind farms, where the vertical kinetic energy

flux dominates. As stated earlier, the blockage effect redistributes energy from the wind farm leading edge into the wind farm, which results in a smaller power density of the first-row wind turbines compared to the other 7D-cases and in a constant power density from approximately row 10. This redistribution is done by a favorable perturbation pressure gradient inside the wind farms and reaches power densities of approximately $1 \, \mathrm{W \, m^{-2}}$ (not shown in Fig. 10). In the wake, the vertical kinetic energy flux at rotor top level drops to zero again, which is consistent with the very low TI in the wake (cf. Fig. 6).

These results show that the power output and the wake of very large wind farms behave very differently compared to small wind farms. The main findings and their implications are summarized in the next section.

## 4    Conclusions

This study investigates wake properties and power output of very large wind farms with different turbine spacings in boundary layers (BL) of different stabilities and heights. Very large wind farms do not only change wind speed and turbulence inten-

sity (TI) at wind turbine level but rather affect several flow quantities inside the entire BL and even above the BL. BL growth, counterclockwise flow deflection inside the BL and clockwise flow deflection above the BL are the main effects that distinguish large from small wind farm flows. Wake lengths of very large wind farms are longer for shallower BLs and smaller turbine spacings, reaching values of more than $100 \, \mathrm{km}$. Thus, very large wind farms in the German Bight have the potential to affect the wind farm performance of neighbouring states such as Denmark or the Netherlands. The wake length in terms of TI is

relatively independent of the wind farm size and is in general much smaller (approximately $20 \, \mathrm{km}$) than the wake length in terms of speed deficit. Longer TI-wakes occur for convective BLs and shorter wakes for stable BLs due to the buoyancy-driven turbulence production or destruction.

For shallow, stable BLs very large wind farms trigger large scale gravity waves in the free atmosphere that cause significant flow blockage, affecting also smaller wind farms that are nearby. Some tuning of the domain height and the boundary conditions

was necessary to capture this phenomenon correctly. Because shallow BLs occur quite frequently in the German Bight, it is an important task to find best practice rules for simulation setups that capture this phenomenon as realistically as possible.

The wind speed recovery inside the wind farms is mainly driven by the turbulent vertical momentum flux but the wind speed recovery in the wake of very large wind farms is mainly driven by the geostrophic pressure gradient force. Thus, it is expected that the wake recovery of very large wind farms rather depends on the ageostrophic wind speed component than on parameters

that affect the turbulent momentum flux such as stability or TI. Further investigations are needed to proof this hypothesis.

The power output of very large wind farms is limited by the available kinetic energy inside the BL and the energy input by the geostrophic pressure gradient. The achieved power density of turbines in the upstream part if the wind farms is significantly affected by the BL height, whereas the power density of the downstream turbines approaches the power density given by the energy input of the geostrophic pressure gradient. Because this power density is only as small as $2 \, \mathrm{W \, m^{-2}}$, the turbine spacing



of very large wind farms should be at least 7 rotor diameters to achieve an acceptable wind farm efficiency. BL growth and wake deflection towards lower pressure tend to increase the power input by the geostrophic pressure gradient, which could have a positive effect on the power output of downstream wind farms.

Overall, the results show that very large wind farms trigger much more complex flow effects than small wind farms do. It will be necessary to consider at least some of these effects in simple wake models in order to accurately predict the power
output of very large wind farms. One of the next research tasks could be to derive empirical rules for predicting the power output of very large wind farms by performing a more systematic and idealized set of simulations.

*Code and data availability.* The PALM code can be accessed under https://palm.muk.uni-hannover.de. PALM input files and plot scripts are available at https://doi.org/10.25835/0004522. Output data are available on request.



## Appendix A:  COSMO-REA6 climatology

This Appendix includes histograms of wind speed (Fig. A1), wind direction (Fig. A2) and boundary layer height for convective boundary layers (Fig. A3) and stable boundary layers (Fig. A4) for a point in $178$ m height at $54°\ 30'$ N, $6°\ 00'$ E, which is located inside Zone 3 in the German Bight. The histograms are obtained from the COSMO-REA6 dataset that contains hourly data from the years 1995 to 2018. The boundary layer height in COSMO-REA6 is defined as the height at which the bulk Richardson number reaches the critical Richardson number, which is 0.22 under convective conditions and 0.33 under

stable conditions (personal communication with Eckhard Kadasch (German Weather Service, Offenbach) on 23.05.2019). The histograms were provided by Thomas Spangehl from the German Weather Service. Note that convective boundary layers occur $59.5$ % of the time ($n = 125088$, Fig. A3) and stable boundary layers occur $40.5$ % of the time ($n = 85247$, Fig. A4).

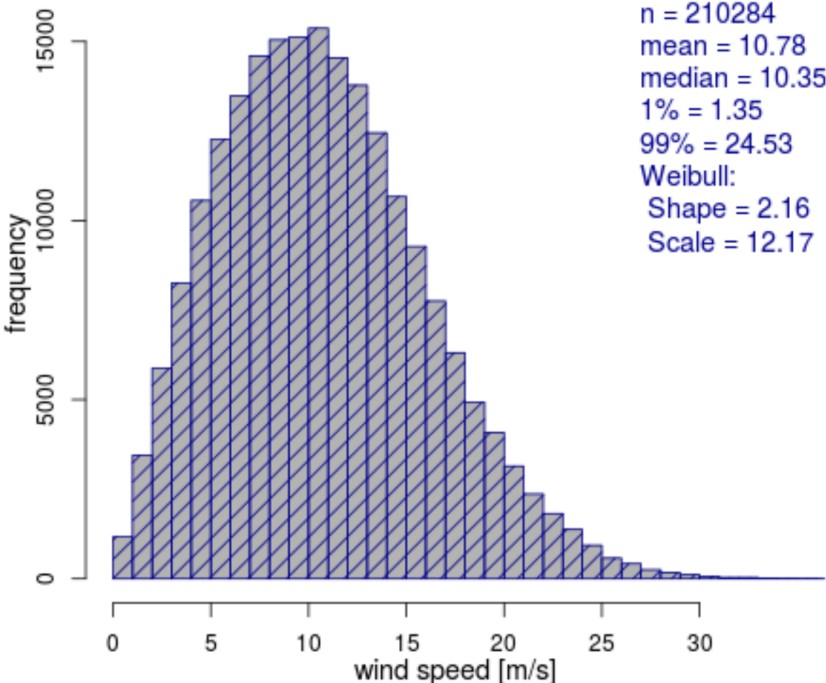

**Figure A1.** Wind speed histogram with total number of samples ($n$), mean wind speed, median wind speed, $1$ % and $99$ % percentile and Weibull shape and scale parameters.

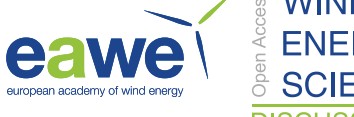

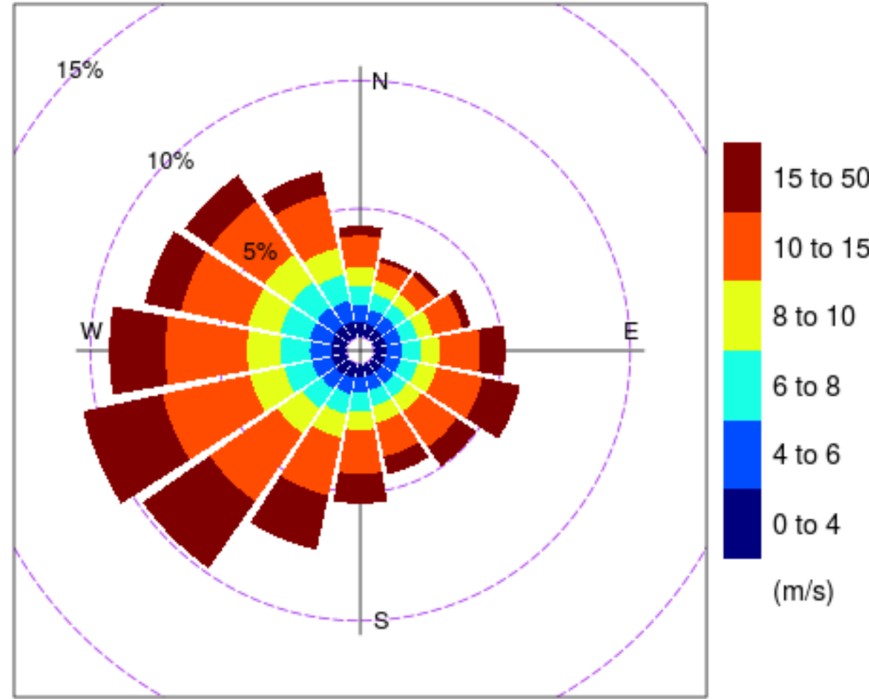

**Figure A2.** Wind direction histogram, wind speed bins are indicated by different colors.

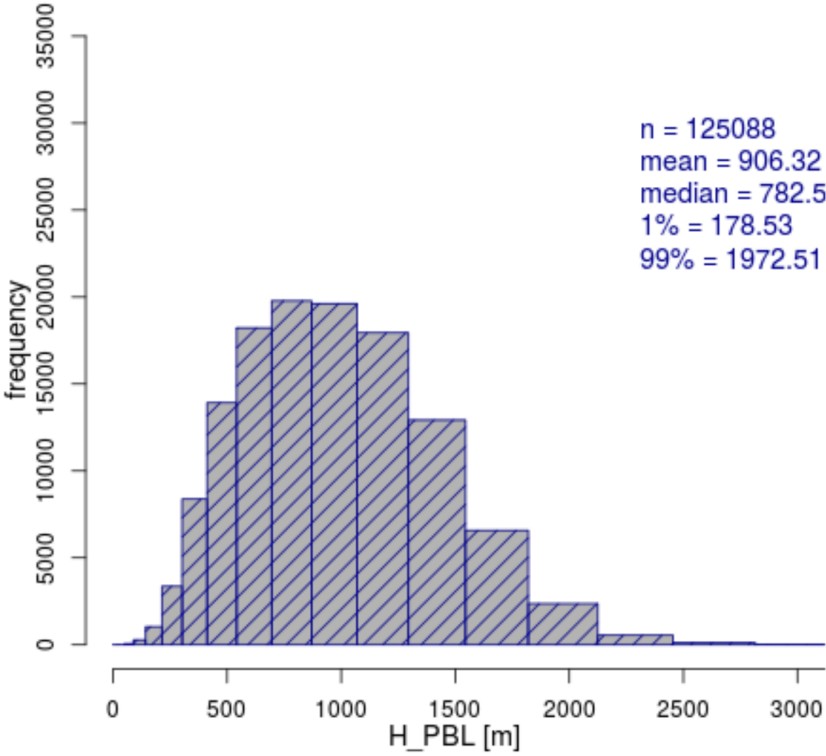

**Figure A3.** Boundary layer height histogram for convective boundary layers (surface temperature greater than 2 m temperature).



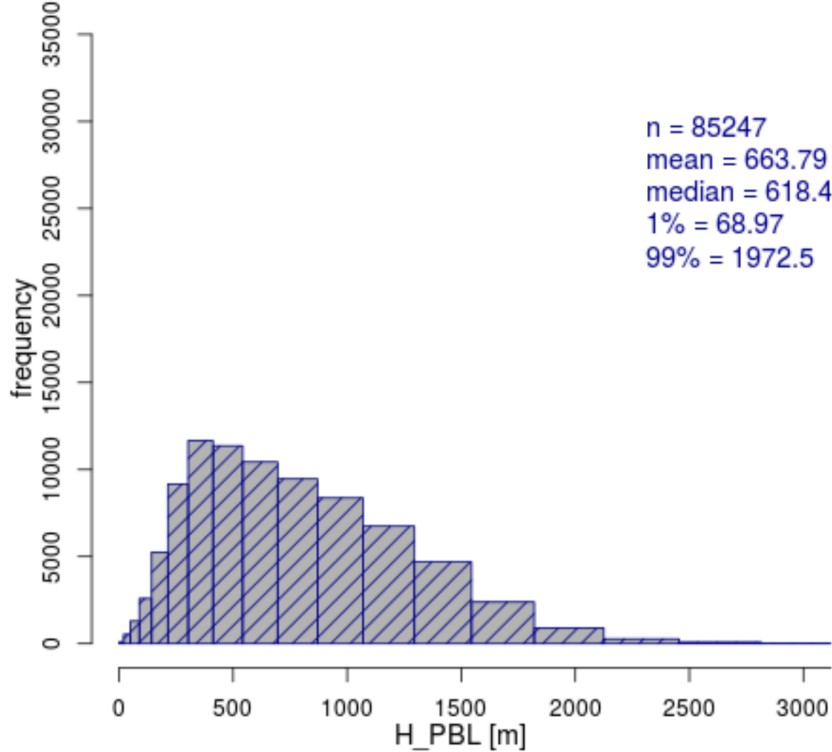

**Figure A4.** Boundary layer height histogram for stable boundary layers (surface temperature smaller than 2 m temperature).



*Author contributions.* O. Maas and S. Raasch selected and designed the simulated cases. O. Maas performed the simulations and wrote the manuscript. Data analysis and manuscript preparation were guided by S. Raasch.

*Competing interests.* The authors declare that they have no conflict of interest.

*Acknowledgements.* The work was supported by the North-German Supercomputing Alliance (HLRN). This work was funded by the Federal Maritime and Hydrographic Agency (BSH), grant number 10044580. The publication of this article was funded by the Open Access Fund of Leibniz Universität Hannover. We thank Thomas Spangehl (German Weather Service) for providing the Figures in the appendix. Special thanks goes to Christopher Mount for English proof reading and to Dries Allaerts for informative discussions about wind farm induced

gravity waves.



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
