# Peer review of "Wake properties and power output of very large wind farms for different meteorological conditions and turbine spacings: A large-eddy simulation case study for the German Bight"

_Wind Energy Science, 2021_

## Referee Comment (RC1)

This paper studies the wake properties and power output of a potential future wind farm scenario in the German Bight by means of massive large-eddy simulations (LES). The simulations cover an area of about 200 by 160 km with a 20 m grid resolution and include several wind farms amounting to a total of up to 2088 wind turbines and representing a total wind farm capacity of up to 31 GW, thereby exceeding any other available LES study by at least an order of magnitude. Five different LES cases are meticulously performed so as to investigate the impact of turbine spacing, boundary-layer height, and atmospheric stability. I believe the LES are well-designed and the resulting LES dataset is quite impressive and potentially very useful for the wind energy community. Further, the authors have presented detailed analyses of the dataset and have found several interesting conclusions. The paper is well written and the figures are very informative. Overall, I believe the research is of good quality and potentially very interesting for the community. However, I do have some major concerns about some of the discussions in the paper. I listed my main concerns below together with other scientific questions and minor/technical comments.

Main concerns:
1. The authors have a tendency to state physical explanations of observed flow behaviour as a fact rather than as a hypothesis or backed-up by specific analysis. While the explanation is often plausible, the authors should more clearly indicate when they are presenting a hypothesis or whether they actually have evidence supporting their claim. For example
   - Line 166: "Due to the self-reinforcing behaviour of this process, the crosswire variations can build up quickly without y-shift, even if the wind farms have a distance of 15 km to the recycling plane." It is unclear to me whether you actually saw this in preliminary calculations or if this is a hypothesis?
   - Line 295: "This BL height dependency occurs because a thicker BL contains more kinetic energy that can be transported down to the wind turbine level by turbulent vertical mixing than a shallow BL." Do you have evidence supporting this hypothesis? If this is something that will be discussed later on, please mention so explicitly.
   - Line 296: "The wake length and speed deficit of small wind farms (e.g. N-1 and N-2) is relatively unaffected by the BL height because the wind farm induced internal boundary layer does not reach the top of the BL." Did you investigate the internal boundary layer development? If not, how can you know that this is the reason? It would be interesting to add IBL development to figure 7 (and extend the analysis to small wind farms as well, see below) to support this hypothesis.
   - Line 314: "The case SBL-300-7D covers several flow features that cannot be seen in the other cases." Are these flow features not visible because they do not occur or because they are smaller? Did you verify quantitatively whether there is any flow deceleration in front of the wind farm in neutral or unstable conditions?
   - Line 418: "Because the mean wind speed inside the BL decreases in the streamwise direction, the IL must be displaced upwards in order to maintain a constant mass flux inside the BL." Note that the mass flux can also be conserved by means of an acceleration above the wind farm (below the IL) or by airflow to the sides. In your results, the IL is displaced upward, but that does not necessarily mean that this will always be the case (e.g., a stronger capping inversion might lead to flow acceleration or flow diversion to the sides). Please rephrase.

- Line 532: "The wind farm efficiencies for the NBL-case are greater because the inflow wind speed in the bulk of the BL is higher for the NBL-case than for the CBL-case." Higher bulk wind speed causing higher efficiencies seems a very plausible explanation, but I'm not sure you can deduce this conclusion from your results with 100% certainty. Maybe rephrase?
- Line 596: "As stated earlier, the power output of infinitely large wind farms is determined by the energy input of the pressure gradient. Hence, the power output of infinitely large wind farms does not depend on the BL height, at least for this idealized setups with a stationary CBL inflow. " Did you actually run simulations of infinitely large wind farms with stationary CBL inflow to confirm this hypothesis?

2. The authors made an effort to cover an enormous area in their simulations, but the bulk of the analysis is based on the large wind farm cluster in zone 3 (e.g., many analyses look at cross-sections at y-120 km). I think it would be useful to include more analyses of the smaller wind farms to be able to contrast the behaviour of "small" and "very large" wind farms, as the difference in flow behaviour between small and very large wind farms appears to be one of the main messages of the paper. For example, figures 5, 7, 8 and 10 focus solely on the very large wind farm cluster and there is no counterpart analysis for one of the small wind farms. Moreover, it is interesting to see in fig 10 that the vertical kinetic energy flux continues to decrease over the wind farm and does not reach a plateau, unlike what was found by Allaerts and Meyers (2017, J. Fluid Mech.). I wonder whether this is because you are looking at much longer wind farms, and therefore a similar analysis for a smaller wind farm would be useful. As a matter of fact, line 615 states that "These results show that the power output and the wake of very large wind farms behave very differently compared to small wind farms." At least in terms of the power output, you haven't shown this in the paper even though the data is available.

3. The authors decided to take a simplified approach to the energy budget analysis, but I believe too many terms are left out to support the discussion. A more detailed or even full energy budget analysis seems warranted.
   a. Line 539: "the extraction of kinetic energy by the wind turbines can be compensated for by two sources of energy: (1) Vertical turbulent flux … and (2) Work done by the geostrophic pressure gradient." This is not entirely correct, as the extraction of kinetic energy can also be compensated by a favourable perturbation pressure gradient induced by gravity waves, which acts on top of the fixed geostrophic pressure gradient.
   b. Line 561: "The wind turbines extract approximately 70% of the total energy input $W\_kef+W\_gpg,wt$." I have a problem with this statement as part of the energy might come from flow deceleration inside the wind farm (i.e., the advective term). By decelerating the flow releases kinetic energy that can be used by the wind turbines or that can be dissipated. The total energy input needs to take this advective source term into account. As shown by Allaerts and Meyers (2017, J. Fluid Mech.), the combination of advective sources and perturbation pressure can be up to 15% of the energy input.
   c. Line 601: "As this influx is proportional to the BL height, much more energy is available for the wind turbines in the case CBL-1400-7D than in the case CBL-700-7D. This results in a higher kinetic energy flux …" I have difficulty with accepting that more energy available automatically leads to higher energy

fluxes without actual data or equations to support this. It would for example be interesting to look at the kinetic energy content of the flow above the wind farm and see how that decreases.

    d. Line 612: "This redistribution is done by a favourable perturbation pressure gradient inside the wind farms and reaches power densities of approximately 1 W/m2 (not shown in Fig. 10)." Energy redistribution due to pressure gradients in the case of wind farm blockage is a major topic nowadays, so showing the actual perturbation pressure gradient contribution could be really helpful in this ongoing debate. I see no reason to exclude it from the figure.

Other scientific questions:

- Abstract and result sections: When talking about power density for very large wind farms, the work by Enrico Antonini comes to mind (see, e.g., Antonini and Caldeira, 2021, papers in Proc. Nat. Sci. Acad. and Applied Energy). How does your work compare to their estimate of the power density limits for large wind farms?
- Abstract and result sections: the clockwise flow deflection above the boundary layer is stated as one of the major findings occuring above very large wind farms and not for small wind farms. However, as mentioned on line 464, the effect might be overestimated due to conservation of mass flux in the FA. Moreover, the sensitivity to the Rayleigh damping layer is said to be out-of-scope, so there is no way of knowing how important this effect is, if it occurs at all. This seems quite a controversial result to me, so I wouldn't highlight it without additional sensitivity analysis or explicitly mentioning the uncertainty about this effect.
- Line 196: "While steady-state turbulence is reached after only a few hours, achieving a steady-state mean flow can take several days." How did you assess stationarity of turbulence and mean-flow quantities? Which quantities did you track and when did you consider it to be steady state (e.g., time rate of change less than 1% per hour)?
- Figure 7 and 8: It is interesting to see that the IL returns to its original height for CBL cases but not for the NBL cases. Is this due to faster wind farm wake recovery in the CBL cases, allowing horizontal convergence?
- Figure 8: $x$ = 120km is identified as the near wake. Could you say how far downstream that is from the wind farm trailing edge, i.e., what is the x-position of the last turbine at this y position?
- Figure 8: The wind speed excess in the FA for NBL-700-5D is greatest at 120km and smaller (but not zero) at 180km. I'm surprised to see that the wind direction at 180km is more ageostrophic than at 120km. This means that there should be some other force at play other than the force balance between pressure gradient and Coriolis force. Or is the flow not in a geostrophic balance due to some kind of inertial oscillation in space (almost like a Rossby wave I suppose, a spatial wave supported by the Coriolis effect?) Can you elaborate? Same goes for NBL-700-7D and SBL case.
- Figure 8: For some cases, the vertical axis includes part of the Rayleigh damping layer. This is particularly visible for the neutral cases. The vertical profiles show non-physical behaviour in the Rayleigh damping layer. I think it would be useful to remind the reader of that by adding a line or a gray zone to indicate the location of the damping layer, or by only showing results below the damping layer.

- Line 505: Why not obtain the reference power for a single turbine using the same inflow profiles of the corresponding case, rather than using the neutral case as a reference for all? Now the reference power does not account for differences in wind shear or wind veer under non-neutral stability conditions, so you are attributing effects due to stability at individual turbine level to wind farm efficiency.
- Line 591: "Figures 10c and d show that a doubling of the BL height has approximately no effect on the energy input by the pressure gradient on the undisturbed inflow." I'd say there is a clear difference: $W_{gpg,BL}$ is 50% higher in CBL-1400-7D than in CBL-700-7D. Are you maybe referring to the sum of $W_{gpg,BL}$ and $W_{gpg,wt}$? Please clarify.
- Line 629: "Some tuning of the domain height and the boundary conditions was necessary to capture this phenomenon correctly." How did you assess whether you capture gravity waves correctly?
- Line 640: "…, the turbine spacing of very large wind farms should be at least 7 rotor diameters to achieve an acceptable wind farm efficiency." What value of wind farm efficiency do you consider acceptable? Isn't this up to the wind farm developer? Similar statements were made in the abstract, so please adjust accordingly.

Minor/technical comments:
- Line 44: For your information, the Lillgrund wind farm has also been simulated with LES by several authors. Consider adding references to, e.g., Churchfield et al. (2012, *AIAA*) and Nilsson et al. (2015, *Wind Energy*) to make this list more complete.
- Line 60: I'm curious as to how much computational resources (in terms of core hours, e.g.) were needed for simulations of this scale. Could you comment on this in the paper?
- Equation 2: The buoyancy term makes use of angular brackets, but the use of these is not defined in the text. Do they indicate horizontal averaging? How do you do that in the main domain which is horizontally not homogeneous?
- Equation 3: The transport equation for internal energy is stated in terms of the potential temperature, while the buoyancy term in equation 2 uses virtual potential temperature. How did you relate these two quantities? Note that neither of these quantities/symbols are defined in the text. Please define all symbols in the text.
- Line 78: You mention that overbars are used to indicate filtered quantities, but the LES equations 1-3 contain no overbars for the velocity components nor for the potential temperature. The notation is inconsistent. Either add overbars consistently in the equation, or mention explicitly that overbars are not shown for these parameters (so only shown for the subgrid-scale stress and heat flux) and that $u_i$ etc. correspond to the filtered quantities.
- Line 142: Please describe what you mean with radiation boundary conditions. Does this simply mean zero velocity gradients and a fixed pressure value, or do you impose more intricate boundary conditions?
- Section 2.4: What boundary condition did you set for the potential temperature at the top of the domain? Please specify in the text.

- Line 225: Basically, subsidence velocity is chosen such that w_sub * Gamma = dtheta_0/dt. Maybe add the equation to clarify how w_sub is set.
- Line 241: "…, as this is the *correct* surface forcing method for SBLs." The word correct sounds quite strong and implies that applying a surface heat flux is incorrect. Rather, Basu et al. show that applying a surface heat flux can only represent weakly stable conditions and the surface forcing method is preferred. I wouldn't go as far as saying that there is a *correct* method for applying boundary conditions for SBLs.
- Section 2.5: This section doesn't add much, so I would try to integrate it elsewhere. Particularly the discussion on how to calculate resolved turbulent fluxes is, as far as I know, standard practice in LES, so I don't think it needs to be mentioned here. I'm not familiar with the term *temporal eddy-correlation method*, is that something you introduced yourself? If not, add references to where this term is used as well.
- Line 335: "Because the wind farms in this study also have a finite size also …" The word "also" appears twice in this sentence, I think one of them can be removed.
- Caption of figure 5: "The forces are normalized … and are horizontally averaged over one turbine spacing along x and y." The main text speaks about horizontal averaging over 4 spacing in x and 2 in y. Please clarify.
- Line 361: Could you state by how much percent the streamwise geostrophic pressure gradient force has increased? It is hard to see in the figure and a quantitative value would be instructive.
- Line 406: "The convective velocity scale is greater in the case CBL-1400-7D than in case CBL-700-7D …" It would be useful to mention the definition of the convective velocity scale here (or add the actual value to table 1) to show why it differs between these two cases.
- Equation 13 and fig 10: Use consistent naming: W_f or W_vkef. There is also a typo on line 561 W_kef.
- Line 637: "The achieved power density of turbines in the upstream part if the wind farms …" This should be "… *of* the wind farms …"

---

## Referee Comment (RC2)

**Reviewer comments to the submission *Wake properties and power output of very large wind farms for different meteorological conditions and turbine spacings: A large-eddy simulation case study for the German Bight**

**Authors:** Oliver Maas and Siegfried Raasch

October 2021

This work investigates the main characteristics of the flow within a cluster of offshore wind parks and their wakes under different atmospheric conditions. In particular, it studies the interaction the wind farm with the surrounding Atmospheric Boundary Layer (ABL) in relation to the atmospheric stability, the ABL height as well as the distance between turbines and the overal size of the park (set by the number of turbines). The study is carried in the German Bight, an area of current and future development for wind projects in the coast of the north sea. Every park in the cluster is outlined following *priority areas* of future development of wind projects. The flow is resolved using LES and the Actuator Disk (AD) is employed to represent the rotors which are placed in a staggered configuration, using two different spacings between rotors that are typical of offshore parks. A study of the power performance of the park in function of the power density available in the different stability and turbine spacing conditions is also presented.

The study is very relevant in the field as it shows the variation of the impact that the ABL stability features has on the flow development within and around very large parks compared to smaller ones. These variations are exemplified by the direction of turning of the wake of the park, the extension and recovery of the velocity and turbulence intensity behind the park as well as the distribution of the available power density in the wind. It presents a detailed analysis of the role played by the atmospheric conditions in the apperance and evolution of the flow within the wake and its consequences in the development of the surrounding ABL flow with focus on the features of the park wakes. It should be noted the very large computational demand to achieve these results: each simulation comprises 2088 wind turbines with mesh sizes ranging from $\sim 7.4 \times 10^9$ to $\sim 1 \times 10^{10}$ cells, which is indeed remarkable (so it will be very informative to reveal the computational expense). I recommend this work for publication but please provide an answer to the following questions and comments:

- L89 "it is possible to prescribe a surface heating ..." is this just a choice or a necessity, in view of the issues with the direct prescription of the surface flux as described in Basu et al. (2008)?

- L94, Could you please briefly comment why the overestimation and how this is fixed? Also, would the use of a different distribution length along the smearing direction help to remedy this (a different value of the standard deviation than $1\Delta x$ in the Gaussian distribution)? I think it is useful to provide these details here (even crucial in the case of the overestimation) as the reference you provide is not in English. Also note that in the archive of Meteorologische Fortbildung at the following link, the issue of the reference you cite appears as from 2015 instead of 2016: `https://www.dwd.de/EN/ourservices/pbfb_verlag_promet/archiv/archiv_promet.html`

- L127 and Table 1: should one assume that 225 deg is the wind direction at hub height obtained with a $u_g$ at direction alpha as shown in the table?

- L133, why have you chosen to use a uniform cell distribution in the horizontal plane? This question arises from the fact that if cell stretching was used away from the turbines, the savings could have been used to achieve a higher refinement either around the rotor and wakes or for the near ground zone, where the mesh requirements for a well-resolved LES are higher.

- L134, why is this resolution enough? In the absence of a numerical or experimental evaluation, please provide an argument. This element is crucial as the recovery of the individual wakes depends on the amount of TKE resolved within and around.

- L134 (and L204), also with regard to the cell resolution, please comment on its the adequacy in terms of the amount of subgrid TKE with respect to the total. In other words, how "well resolved" is the LES, especially for the wake region? L204, please describe how the subgrid convergence study demonstrated that 20 m was the adequate cell resolution in the horizontal directions.

- L165 and results section: it is mentioned that blockage from the parks is observed to reach the recycling plane, did the authors observe any dependence of the blockage with respect to the size of the domain (regardless of the fact that the boundary conditions at the sides are periodic)?

- L168, it's not totally clear how the height limitation set to the recycling of turbulence helps to avoid the increase in the BL height, is the subsidence mentioned in L90 not sufficient? However, it is clear that this region represents only a small portion of the total domain length, albeit still large (10 km).

- L194, please clarify the usage of the concept "steady state". It seems that in some instances the authors use these words when referring to a flow that has achieved *statistical convergence* such as in a developed BL which could be confusing with a flow that is indeed steady-state and as such, lacks transient features such as those reproduced by LES.

- L230,235, could the issues pointed at by Basu et al. (2008), such as those related to the estimation of the friction velocity, be added to these arguments?

- L378, is it possible to provide an example (or a threshold) of a park size where, following these arguments, the turning would occur in the clockwise direction?

- In Figs 5 and 6, scale is inverse with respect to the other (red to green and green to red), please maintain consistency to denote low to large values in scaling. Furthermore, it is rather common to see blue to red to depict small to large values (e.g. subfigure (f) in figures 4, 6 and 7), so to use blue in the middle can be somewhat disconcerting. Is this perhaps a choice to highlight the wakes behind the clusters?

- L386, the TKE does consider the subgrid scales. Since the resolution is relatively coarse to resolve the flow around the rotors, a sizable part of the TKE is potentially found within the subgrid scales. Please provide an argument regarding why only resolved scales should be considered in this analysis.

- Figure 7, label in vertical scale should be $z$.

- L505, how is the power computed for each turbine?

- L608, it is remarked that for the SBL-300-7D case, the $W_{gpg,wt}$ component is dominant but in the Fig. 10(e) it seems that $W_{vkef}$ is about the same value throughout except only within the two gaps between farms.

**Technical corrections**

- L121, typo, replace "can not" for "cannot"

- L162: I suggest to remove first comma

- L308, typo, it should be "effect" (it says affect)

- L637, typo, "of" instead of "if"

---

## Author Comment (AC1)

We first like to thank the reviewer for the comprehensive, fair and constructive comments. Please find our answers to the comments below. We marked the reviewers comments blue and our answers black.

Main concerns:

1. The authors have a tendency to state physical explanations of observed flow behaviour as a fact rather than as a hypothesis or backed-up by specific analysis. While the explanation is often plausible, the authors should more clearly indicate when they are presenting a hypothesis or whether they actually have evidence supporting their claim.

Checking the mentioned statements revealed that some of them are not correct and have to be revised.

For example

- Line 166: "Due to the self-reinforcing behaviour of this process, the crosswise variations can build up quickly without y-shift, even if the wind farms have a distance of 15 km to the recycling plane." It is unclear to me whether you actually saw this in preliminary calculations or if this is a hypothesis?

We added a sentence on that:

"Test simulations without y-shift showed that, due to the self-reinforcing behavior of this process, the crosswise variations of wind speed can build up to +- 2%."

- Line 295: "This BL height dependency occurs because a thicker BL contains more kinetic energy that can be transported down to the wind turbine level by turbulent vertical mixing than a shallow BL." Do you have evidence supporting this hypothesis? If this is something that will be discussed later on, please mention so explicitly.

Yes, the vertical kinetic energy flux is greater for the thicker BL, as can be seen in Fig. 10.

Rephrase: "This BL height dependency occurs because the turbulent vertical kinetic energy flux is greater for the case with the thicker BL, cf. Fig. 10 c and d."

- Line 296: "The wake length and speed deficit of small wind farms (e.g. N-1 and N-2) is relatively unaffected by the BL height because the wind farm induced internal boundary layer does not reach the top of the BL." Did you investigate the internal

boundary layer development? If not, how can you know that this is the reason? It
would be interesting to add IBL development to figure 7 (and extend the analysis
to small wind farms as well, see below) to support this hypothesis.

We now included IBL-lines in Figure 7 and added a new figure with vertical cross sections through
a small wind farm (y = 50 km), that shows the IBL behaviour (Figures shown later under reviewers
point 2). We rephrased line 296 ff like this:

"The wind speed deficit and the wake length of small wind farms (e.g. N-1, N-2 and N-3) is
relatively unaffected by the BL height because the wind farm induced internal BL does not reach
the inversion layer (NBL cases) or only reaches it several 10 km downstream of the wind farm
trailing edge (CBL cases), cf. Fig. 7.5_new. Consequently, the BL height only affects further wind
speed recovery (e.g. to 9.5 m/s) in the far wake of small wind farms. For example, the wind speed
recovery from 9 to 9.5 m/s in the wake of N-2 takes longer for the case CBL-700-7D (~40 km) than
for the case CBL-1400-7D (~15 km), cf. Fig 4 c and d."

• Line 314: "The case SBL-300-7D covers several flow features that cannot be seen
in the other cases." Are these flow features not visible because they do not occur
or because they are smaller? Did you verify quantitatively whether there is any
flow deceleration in front of the wind farm in neutral or unstable conditions?

They are smaller. We rephrased this part:
"The case SBL-300-7D covers several flow features that are not as significant in the other cases."

In line 320 we added information about the flow blockage in the other cases:

"At a distance of 2.5 D upstream of the first wind turbine row of the wind farms in Zone 3 the wind
speed is reduced by approximately 10% relative to the inflow wind speed. For all other cases the
speed reduction is approximately 2%."

We carefully checked the Froude number (according to the definition in Wu and Porté-Agel, 2017)
for all cases and found that this parameter is not suitable for predicting flow blockage (at least for
the 5 cases that we investigated). The flow is subcritical (and thus indicating flow blockage,
according to Wu and Porté-Agel) only in the case CBL-1400-7D, but in this case the flow blockage
is very small. Additionally we mixed up subcritical with supercritical, probably due to a mistake in
Table 2 of Wu and Porté-Agel (2017) (they mixed up the Froude numbers). So we rephrased the text
from line 320 to 323 as:

"Wu and Porté-Agel (2017) reported 11 % speed reduction 2.5 D upstream of the first turbine row
of a 20 km long wind farm in a CNBL with a FA stratification of $\Gamma$ = +5 K km$^{-1}$ . However, for $\Gamma$ =
+1 K km$^{-1}$ they reported a speed reduction of only 1.2 %, because the flow is supercritical (Froude
number F r < 1). Using the same definition2 as in Wu and Porté-Agel (2017), the Froude number in

the case SBL-300-7D is F r = 1.47, indicating a supercritical flow. This should, according to the reasoning of Wu and Porté-Agel, result in a weak flow blockage, which does not correspond to the significant flow blockage observed in the case SBL-300-7D. The only case that is subcritical (and should thus show significant flow blockage) is CBL-1400-7D (Fr=0.81), but in this case the flow blockage is only very weak. Hence, for the cases that are investigated in this study, the Froude number, as defined by Wu and Porté-Agel, is not an appropriate parameter for predicting flow blockage."

- Line 418: "Because the mean wind speed inside the BL decreases in the streamwise direction, the IL must be displaced upwards in order to maintain a constant mass flux inside the BL." Note that the mass flux can also be conserved by means of an acceleration above the wind farm (below the IL) or by airflow to the sides. In your results, the IL is displaced upward, but that does not necessarily mean that this will always be the case (e.g., a stronger capping inversion might lead to flow acceleration or flow diversion to the sides). Please rephrase.

Yes, that is not precise enough, right. But if we write 'mean wind speed inside the BL', than that already includes the effect of acceleration above wind farm (below IL). We rephrased this to:

"The IL displacement is a result of the reduced wind speed in the bulk of the BL: To obtain a divergence-free flow inside the BL, the wind speed reduction (streamwise convergence) is compensated by vertical divergence (IL displacement) and crosswise divergence (flow around the wind farms)."

- Line 532: "The wind farm efficiencies for the NBL-case are greater because the inflow wind speed in the bulk of the BL is higher for the NBL-case than for the CBL-case." Higher bulk wind speed causing higher efficiencies seems a very plausible explanation, but I'm not sure you can deduce this conclusion from your results with 100% certainty. Maybe rephrase?

Yes, this is a hypothesis and should not be written as a statement. Rephrase (from line 531):

"A comparison between the cases NBL-700-7D and CBL-700-7D shows that greater wind farm efficiencies are obtained for the NBL, although better efficiencies are expected for the CBL due to the better vertical mixing. Comparing the wind speed profiles of this cases (cf. Fig. 3) shows that the inflow wind speed in the bulk of the BL is higher for the NBL than for the CBL, which is probably the reason for the higher wind farm efficiencies."

- Line 596: "As stated earlier, the power output of infinitely large wind farms is

> determined by the energy input of the pressure gradient. Hence, the power output
>
> of infinitely large wind farms does not depend on the BL height, at least for this
>
> idealized setups with a stationary CBL inflow. " Did you actually run simulations of
>
> infinitely large wind farms with stationary CBL inflow to confirm this hypothesis?

No, we only expect this behaviour. Rephrase:
"As stated earlier, the power output of infinitely large wind farms is determined by the energy input of the pressure gradient, which does not depend on the BL height. Hence, the power output of infinitely large wind farms is expected not to depend on the BL height, at least for this idealized setups with a stationary CBL inflow."

> 2. The authors made an effort to cover an enormous area in their simulations, but the
>
> bulk of the analysis is based on the large wind farm cluster in zone 3 (e.g., many
>
> analyses look at cross-sections at y-120 km). I think it would be useful to include more
>
> analyses of the smaller wind farms to be able to contrast the behaviour of "small" and
>
> "very large" wind farms, as the difference in flow behaviour between small and very
>
> large wind farms appears to be one of the main messages of the paper. For example,
>
> figures 5, 7, 8 and 10 focus solely on the very large wind farm cluster and there is no
>
> counterpart analysis for one of the small wind farms. Moreover, it is interesting to see
>
> in fig 10 that the vertical kinetic energy flux continues to decrease over the wind farm
>
> and does not reach a plateau, unlike what was found by Allaerts and Meyers (2017, J.
>
> Fluid Mech.). I wonder whether this is because you are looking at much longer wind
>
> farms, and therefore a similar analysis for a smaller wind farm would be useful. As a
>
> matter of fact, line 615 states that "These results show that the power output and the
>
> wake of very large wind farms behave very differently compared to small wind farms."
>
> At least in terms of the power output, you haven't shown this in the paper even
>
> though the data is available.

This is a legitimate objection and we agree that it is a good idea to show more results for the small wind farms. However, the paper is already very long and includes many large figures. Showing figures 5, 7, 8 and 10  additionally for small wind farms and discussing these figures would lead to at least 50 pages in total, which is too much in our opinion. As a compromise, we decided to add a figure like Fig. 7 (see below), that shows the BL development for a small wind farm (N-2, y=50 km). We think that this additional material on small wind farms has to be sufficient, also because the main focus of this paper is on the effects of very large wind farms. The investigation of

differences between small and large wind farms will definitely be part of a follow up study that we are currently working on.

We added a paragraph which discusses the new figure in the revised manuscript

3. The authors decided to take a simplified approach to the energy budget analysis, but I believe too many terms are left out to support the discussion. A more detailed or even full energy budget analysis seems warranted.

    a. Line 539: "the extraction of kinetic energy by the wind turbines can be compensated for by two sources of energy: (1) Vertical turbulent flux ... and (2) Work done by the geostrophic pressure gradient." This is not entirely correct, as the extraction of kinetic energy can also be compensated by a favourable perturbation pressure gradient induced by gravity waves, which acts on top of the fixed geostrophic pressure gradient.

This is true, we added it in the list:

 The extraction of kinetic energy by the wind turbines can be compensated for by three sources of energy:

"

    1. Vertical turbulent flux of kinetic energy at rotor top level, $W\_f$

2. Work done by the geostrophic pressure gradient on the flow below rotor top level, W_gpg,wt

3. Work done by the perturbation pressure gradient on the flow below rotor top level, W_ppg,wt

"

We also added a equation for W_ppg and a graph for the work done by the perturbation pressure gradient in Fig 10 / now Fig. 11 (see further below).

The power density of the energy input by the perturbation pressure ($p^*$) gradient on the flow below rotor top level $z_t = 270$ m is calculated as:

$$W_{ppg,wt}(x) = -\int\limits_{z=0}^{z_t} \frac{\partial p^*(z)}{\partial x}\overline{u}(z) + \frac{\partial p^*(z)}{\partial y}\overline{v}(z)dz|_{y=120 \text{ km}}. \tag{16}$$

b. Line 561: "The wind turbines extract approximately 70% of the total energy input W_kef+W_gpg,wt." I have a problem with this statement as part of the energy might come from flow deceleration inside the wind farm (i.e., the advective term). By decelerating the flow releases kinetic energy that can be used by the wind turbines or that can be dissipated. The total energy input needs to take this advective source term into account. As shown by Allaerts and Meyers (2017, J. Fluid Mech.), the combination of advective sources and perturbation pressure can be up to 15% of the energy input.

Yes that is true. We added a note on this in the comparison with the infinite wind farms:

"The wind turbines extract approximately 70 % of the total energy input Wkef + Wgpg,wt + *W_ppg,wt*, which is a relatively large value. Johnstone and Coleman (2012) and Abkar and Porté-Agel (2014), who analyzed the energy budgets for an infinite wind farm in a NBL, reported that 35 % and 45 %, respectively, of the energy input by the geostrophic pressure gradient is extracted by the wind turbines. The difference could be explained by the fact that, for the finite wind farms in this study, additional energy is provided to the turbines by the divergence of horizontal advection of kinetic energy (flow deceleration). Additionally, the differences could be explained by the low Reynolds number of Re = 1000 in the simulations of Johnstone and Coleman (2012) and the higher roughness length of z0 = 0.1 m in the simulations of Abkar and Porté-Agel (2014)."

Except for the case SBL-300-7D, the power input by the perturbation pressure averaged over the entire farm length is approximately zero. Thus the 70 % statement is still correct for the NBL case discussed here.

c. Line 601: "As this influx is proportional to the BL height, much more energy is

available for the wind turbines in the case CBL-1400-7D than in the case CBL-

700-7D. This results in a higher kinetic energy flux ..." I have difficulty with

accepting that more energy available automatically leads to higher energd.fluxes without actual data or equations to support this. It would for example

be interesting to look at the kinetic energy content of the flow above the wind

farm and see how that decreases.

We deleted this reasoning and just state that a thicker BL results in a greater vertical kinetic energy flux and turbine power densities.

"However, for very large, but finite-size wind farms, as in this study, the power output depends significantly on the BL height, as it is shown in Fig. 10c and d. The vertical kinetic energy flux is greater and decays slower for the thicker BL (CBL-1400-7D), resulting in higher turbine power densities."

Line 612: "This redistribution is done by a favourable perturbation pressure

gradient inside the wind farms and reaches power densities of approximately

1 W/m2 (not shown in Fig. 10)." Energy redistribution due to pressure

gradients in the case of wind farm blockage is a major topic nowadays, so

showing the actual perturbation pressure gradient contribution could be really

helpful in this ongoing debate. I see no reason to exclude it from the figure.

We have included the perturbation pressure contribution in Fig. 10 (now Fig. 11, see below).

[Figure]

Other scientific questions:

- Abstract and result sections: When talking about power density for very large wind farms, the work by Enrico Antonini comes to mind (see, e.g., Antonini and Caldeira, 2021, papers in Proc. Nat. Sci. Acad. and Applied Energy). How does your work compare to their estimate of the power density limits for large wind farms?

We thank the reviewer for mentioning this article, which we haven't been aware of. Antoninis results show a power density of 1-1.5 W/m² for G=12 m/s and lat = 46.1°, which is consistent with our findings. We refered to this article in line 586ff in the results section:

"The energy input by the pressure gradient (Wgpg,wt + Wgpg,BL) achieves power densities of only $1 - 2$ W m−2, which is consistent with the geophysical limits to power densities of large wind farms found by Antonini and Caldeira (2021), who reported approximately 1.5 W m−2 for a latitude of 46 ◦ and a geostrophic wind speed of 12 ms−1 . This power density is much smaller than the power density achieved by the first-row wind turbines. As the case NBL-700-5D shows (Fig. 11b), a reduction of the turbine spacing from s = 7 D to s = 5 D approximately results in a doubling of the power density of the first-row wind turbines (from 4.5 W m−2 to 8.5 W m−2 ), but the power density of the last row wind turbines is as low as for s = 7 D. "

Abstract and result sections: the clockwise flow deflection above the boundary layer is stated as one of the major findings occuring above very large wind farms and not for small wind farms. However, as mentioned on line 464, the effect might be overestimated due to conservation of mass flux in the FA. Moreover, the sensitivity to the Rayleigh damping layer is said to be out-of-scope, so there is no way of knowing how important this effect is, if it occurs at all. This seems quite a controversial result to me, so I wouldn't highlight it without additional sensitivity analysis or explicitly mentioning the uncertainty about this effect.

In line 477 we state that this effect (as well as the speed up above the BL) might be overestimated. Our current investigations show that this effect also occurs if the vertical distance to the rayleigh-damping layer is much greater. So we are confident, that this effect is real.

Line 196: "While steady-state turbulence is reached after only a few hours, achieving a steady-state mean flow can take several days." How did you assess stationarity of turbulence and mean-flow quantities? Which quantities did you track and when did you consider it to be steady state (e.g., time rate of change less

We added the following lines to clarify how we define steady-state:

"While steady-state turbulence is reached after only a few hours, achieving a steady-state mean flow can take several days. Here, we declare the mean flow as steady if the oscillation amplitude of the hub height mean wind speed is less than 0.5 % and declare the turbulence as steady if the change in friction velocity is less than 2 % in 4 h."

Figure 7 and 8: It is interesting to see that the IL returns to its original height for

CBL cases but not for the NBL cases. Is this due to faster wind farm wake recovery

in the CBL cases, allowing horizontal convergence?

With this setup and the available data we can only guess why this happens so that we decided to not further comment in this effect. It is difficult to seperate different mechanisms, that could be responsible for this effect. For example the heterogeneity in y-direction makes an analysis difficult. In the follow-up study we use a more idealized setup that will allow us to investigate this effect much better.

Figure 8: x = 120km is identified as the near wake. Could you say how far

downstream that is from the wind farm trailing edge, i.e., what is the x-position of

the last turbine at this y position?

We added this information in the figure caption:

"The wind farm trailing edge is located at $x=108~\unit{km}$"

Figure 8: The wind speed excess in the FA for NBL-700-5D is greatest at 120km and

smaller (but not zero) at 180km. I'm surprised to see that the wind direction at

180km is more ageostrophic than at 120km. This means that there should be some

other force at play other than the force balance between pressure gradient and

Coriolis force. Or is the flow not in a geostrophic balance due to some kind of

inertial oscillation in space (almost like a Rossby wave I suppose, a spatial wave

supported by the Coriolis effect?) Can you elaborate? Same goes for NBL-700-7D

and SBL case.

Yes exactly, it is some kind of inertial oscillation in space. We added a sentence on this in line 475:

"Note that the highest wind speed excess occurs at $x=120~\unit{km}$ but the highest deflection angle occurs at $x=180~\unit{km}$. This effect can be interpreted as an inertia oscillation in space (along $x$), with the deflection angle being $90^\circ$ phase shifted relative to the wind speed excess."

Figure 8: For some cases, the vertical axis includes part of the Rayleigh damping layer. This is particularly visible for the neutral cases. The vertical profiles show non-physical behaviour in the Rayleigh damping layer. I think it would be useful to remind the reader of that by adding a line or a gray zone to indicate the location of the damping layer, or by only showing results below the damping layer.

We now set the upper limit of the z axis to below the damping layer.

Line 505: Why not obtain the reference power for a single turbine using the same inflow profiles of the corresponding case, rather than using the neutral case as a reference for all? Now the reference power does not account for differences in wind shear or wind veer under non-neutral stability conditions, so you are attributing effects due to stability at individual turbine level to wind farm efficiency.

We now also ran simulations for CBL 700 CBL, 1400 and SBL 300 cases to obtain reference powers for these cases. Only the SBL case shows a significantly different reference power and thus higher wind farm efficiency (see table below). We added the numbers in Table 2 and in the text.

Reference power (corrected for power overestimation):

NBL: 12.56 MW

CBL 700: 12.51 MW

CBL 1400: 12.53 MW

SBL: 11.45 MW

--> Reference power is smaller for SBL case. This results in higher wind farm efficiency:

| case | Wind farm efficiency (old/new) | |
|---|---|---|
| | N-1 | Zone 3 |
| NBL-700-7D | 0.87/0.87 | 0.58/0.58 |
| NBL-700-5D | 0.77/0.77 | 0.41/0.41 |
| CBL-700-7D | 0.86/0.86 | 0.54/0.54 |
| CBL-1400-7D | 0.86/0.88 | 0.63/0.64 |
| SBL-300-7D | 0.61/66 | 0.42/0.46 |

We also corrected the values in the text. The overall message and conclusion stays the same.

Line 591: "Figures 10c and d show that a doubling of the BL height has approximately no effect on the energy input by the pressure gradient on the undisturbed inflow." I'd say there is a clear difference: W_gpg,BL is 50% higher in CBL-1400-7D than in CBL-700-7D. Are you maybe referring to the sum of W_gpg,BL and W_gpg,wt? Please clarify.

Yes, we meant the sum.

"Figure 10c and d show that a doubling of the BL height has approximately no effect on the energy input by the geostrophic pressure gradient ($W_{gpg,wt} + W_{gpg,BL}$) on the undisturbed inflow."

Line 629: "Some tuning of the domain height and the boundary conditions was necessary to capture this phenomenon correctly." How did you assess whether you capture gravity waves correctly?

L441 - 449

We rephrased that: "Some tuning of the domain height and the boundary conditions was

necessary to obtain stable simulation results." We did not investigate the interaction of waves with the top and inflow boundary in detail but in the following sentence we highlight that it is an important task to find best practice rules for simulation setups that capture this phenomenon as realistically as possible.

Line 640: "..., the turbine spacing of very large wind farms should be at least 7 rotor diameters to achieve an acceptable wind farm efficiency." What value of wind farm efficiency do you consider acceptable? Isn't this up to the wind farm developer? Similar statements were made in the abstract, so please adjust accordingly.

That is true. We rephrase this to a more general statement:
"Because this power density is only as small as 2 W m−2, high wind farm efficiencies can only be achieved by large turbine spacings."

Minor/technical••••••comments:

Line 44: For your information, the Lillgrund wind farm has also been simulated with

LES by several authors. Consider adding references to, e.g., Churchfield et al. (2012,

AIAA) and Nilsson et al. (2015, Wind Energy) to make this list more complete.

We added the mentioned articles and thank the reviewer for this hint.

Line 60: I'm curious as to how much computational resources (in terms of core hours,

e.g.) were needed for simulations of this scale. Could you comment on this in the

paper?

We added information about number of cores and computing time, also due to a comment of the other reviewer.

"The simulations were carried out on 5120 cores on one of the supercomputers of the North-German Supercomputing Alliance (HLRN). A simulation required a Wallclock time of 25 to 50 h."

Equation 2: The buoyancy term makes use of angular brackets, but the use of these is

not defined in the text. Do they indicate horizontal averaging? How do you do that in

the main domain which is horizontally not homogeneous?

We added the information that angular brackets indicate horizontal averaging. This rule is also applied for the main domain. The (in our opinion worse) alternative would be to use a fixed (constant in time) reference temperature profile.

Equation 3: The transport equation for internal energy is stated in terms of the

potential temperature, while the buoyancy term in equation 2 uses virtual potential

temperature. How did you relate these two quantities? Note that neither of these

quantities/symbols are defined in the text. Please define all symbols in the text.

Since humidity is not considered in the simulations, theta = theta_v, so we changed theta_v to theta in equation 2. We also added a definition in the text.

Line 78: You mention that overbars are used to indicate filtered quantities, but the LES

equations 1-3 contain no overbars for the velocity components nor for the potential

temperature. The notation is inconsistent. Either add overbars consistently in the

equation, or mention explicitly that overbars are not shown for these parameters (so

only shown for the subgrid-scale stress and heat flux) and that u_i etc. correspond to

the filtered quantities.

We now mentioned that these are filtered quantities and did not add overbars.

Line 142: Please describe what you mean with radiation boundary conditions. Does

this simply mean zero velocity gradients and a fixed pressure value, or do you impose

more intricate boundary conditions?

We added a reference:

"Details about the radiation BC can be found in Miller and Thorpe (1981) and Orlanski (1976)."

The flow field is advected past the boundary with the maximum allowed speed according to the CFL criterium.

Section 2.4: What boundary condition did you set for the potential temperature at the

top of the domain? Please specify in the text.

We added this info together with the pressure BC in line  145:

"At the domain top a Neumann boundary condition is set for the perturbation pressure and the vertical potential temperature gradient is kept constant."

Line 225: Basically, subsidence velocity is chosen such that w_sub * Gamma =

dtheta_0/dt. Maybe add the equation to clarify how w_sub is set.

We added this equation.

Line 241: "..., as this is the correct surface forcing method for SBLs." The word correct

sounds quite strong and implies that applying a surface heat flux is incorrect. Rather,

Basu et al. show that applying a surface heat flux can only represent weakly stable

conditions and the surface forcing method is preferred. I wouldn't go as far as saying

that there is a correct method for applying boundary conditions for SBLs.

We rephrased that to:

 "A Dirichlet-condition is applied for the surface temperature, because prescribing a surface heat flux can lead to unphysical results (Basu et al., 2008)"

Section 2.5: This section doesn't add much, so I would try to integrate it elsewhere.

Particularly the discussion on how to calculate resolved turbulent fluxes is, as far as I

know, standard practice in LES, so I don't think it needs to be mentioned here. I'm not

familiar with the term temporal eddy-correlation method, is that something you

introduced yourself? If not, add references to where this term is used as well.

We think that it is a good idea to clearly show how the fluxes and v_h are calculated and would like to keep this small section. We skipped the term "temporal", because "eddy correlation method" usually means correlation of timeseries and not correlation of 2d or 3d spatial fields.

Line 335: "Because the wind farms in this study also have a finite size also ..." The word "also" appears twice in this sentence, I think one of them can be removed.

Yes, is corrected.

Caption of figure 5: "The forces are normalized ... and are horizontally averaged over one turbine spacing along x and y." The main text speaks about horizontal averaging over 4 spacing in x and 2 in y. Please clarify.

One spacing is correct. We changed it in the main text.

Line 361: Could you state by how much percent the streamwise geostrophic pressure gradient force has increased? It is hard to see in the figure and a quantitative value would be instructive.

It's 80%, We added it.

Line 406: "The convective velocity scale is greater in the case CBL-1400-7D than in case CBL-700-7D ..." It would be useful to mention the definition of the convective velocity scale here (or add the actual value to table 1) to show why it differs between these two cases.

We added the definition in the text.

Equation 13 and fig 10: Use consistent naming: W_f or W_vkef. There is also a typo on line 561 W_kef.

We changed all terms to W_{vkef}

We corrected that.

---

## Author Comment (AC2)

We first like to thank the reviewer for the comprehensive, fair and constructive comments. Please find our answers to the comments below. We marked the reviewers comments blue and our answers black.

It should be noted the very large computational demand to achieve these results: each simulation comprises 2088 wind turbines with mesh sizes ranging from ~ 7.4 × 10⁹ to ~ 1 × 10¹⁰ cells, which is indeed remarkable (so it will be very informative to reveal the computational expense).

In line 134 we added a sentence on that:
"The simulations were carried out on 5120 cores on one of the supercomputers of the North-German Supercomputing Alliance (HLRN). A simulation required a Wallclock time of 25 to 50 h."

I recommend this work for publication but please
provide an answer to the following questions and comments:

L89 "it is possible to prescribe a surface heating . . . " is this just a choice or a necessity, in view of the issues with the direct prescription of the surface flux as described in Basu et al. (2008)?

The importance of prescribing a surface heating rate instead of a heat flux for SBLs (and CBLs) is highlighted later in sections 2.4.2 and 2.4.3.
In section 2.4.3 (SBL) we rephrased line 241 to:
"A Dirichlet-condition is applied for the surface temperature, because prescribing a surface heat flux can lead to unphysical results (Basu et al., 2008)"

L94, Could you please briefly comment why the overestimation and how this is fixed? Also, would the use of a different distribution length along the smearing direction help to remedy this (a different value of the standard deviation than 1Δx in the Gaussian distribution)? I think it is useful to provide these details here (even crucial in the case of the overestimation) as the reference you provide is not in English. Also note that in the archive of Meteorologische Fortbildung at the following link, the issue of the reference you cite appears as from 2015 instead of 2016: https://www.dwd.de/EN/ourservices/pbfb_verlag_promet/archiv/archiv_promet.html

Yes, we kept this part short. We have now added some information and changed the year to 2015:

"The ADM is described in detail by Steinfeld et al.(2015) and Wu and Porté-Agel (2011). To avoid numerical instabilities, the disc element forces are distributed to the neighboring grid points by a three-dimensional Gaussian smearing kernel, which is approximated by computationally less expensive 4th-order polynomial. The smearing kernel has a default radius of 2Δx, reaching approximately 78 grid points. The otherwise two-dimensional actuator disc is enlarged in the axial and radial direction by the smearing, resulting in a power overestimation of 26.8 %. The power overestimation can be reduced to 12.5 % by setting the kernel radius to 1Δx, reaching approximately 10 grid points, without any numerical instabilities. The thrust coefficient is overestimated by 2% for 2Δx and underestimated by 4% for 1Δx. As a compromise, the smearing kernel radius is set to 1Δx for this study. The wind turbine power output is corrected for the power overestimation by a factor of 1/1.125 before entering the wind farm power output analysis."

L127 and Table 1: should one assume that 225 deg is the wind direction at hub height obtained

with a ug at direction alpha as shown in the table?

Yes, it is hub height wind direction. We added that:

"The wind direction at hub height is set to 225° by tuning the geostrophic wind direction \alpha appropriately (cf. Table 1). Southwest-wind is one of the most common wind directions in the German Bight."

L133, why have you chosen to use a uniform cell distribution in the horizontal plane? This question arises from the fact that if cell stretching was used away from the turbines, the savings could have been used to achieve a higher refinement either around the rotor and wakes or for the near ground zone, where the mesh requirements for a well-resolved LES are higher.

PALM allows only for a vertical stretching. In the horizontal directions the cells have to be uniform. The nesting feauture of PALM could have been used for sparing resources, but we decided against this option in order to avoid any numerical effects on our results that might be introduced by the nesting. Thus we decided to use this uniform grid with 20 m grid spacing.

L134, why is this resolution enough? In the absence of a numerical or experimental evaluation, please provide an argument. This element is crucial as the recovery of the individual wakes de- pends on the amount of TKE resolved within and around.

There is a numerical evaluation in Steinfeld (2015) that shows that 5D behind the rotor (which is the smallest turbine spacing used in our study and due to the staggered configuration the first wake-turbine interaction happens at 10 D) the wind speed profiles for a grid spacing of below 10 grid points per rotor diameter are the same as for finer grid spacings. They used the 5MW NREL turbine with D=126m and results with 16 m grid spacing were still good at 5D, Fig. 4-7.

We added this information:

"This grid spacing yields a density of 12 grid points per rotor diameter, which is enough to resolve the most relevant eddies inside the wind turbine wakes. As Steinfeld (2015) showed, even 8 grid points per rotor diameter are sufficient to obtain a converged result for the mean wind speed profiles 5D behind the turbine."

L134 (and L204), also with regard to the cell resolution, please comment on its the adequacy in terms of the amount of subgrid TKE with respect to the total. In other words, how "well resolved" is the LES, especially for the wake region? L204, please describe how the subgrid convergence study demonstrated that 20 m was the adequate cell resolution in the horizontal directions.

The ratio of SGS-TKE to total TKE and SGS-momentum flux to total momentum flux is smaller than 10% for the SBL precursor run.

We added this information:

"Test simulations with 10 m and 20 m grid spacing showed that a grid spacing of 20 m is sufficient, if this SGS-model is used (less than 1 % difference in wind speed maximum and less than 5 % difference in BL height), whereas the results are more grid spacing sensitive (2 % difference in wind speed maximum and 20 % difference in BL height) if the standard-SGS model of PALM is used. For the SBL precursor run the ratio of SGS-TKE to total TKE and SGS-momentum flux to total momentum flux is smaller than 10%, except for the lowest grid point."

For SGS-TKE in the wake see the comment on line 386.

L165 and results section: it is mentioned that blockage from the parks is observed to reach the recycling plane, did the authors observe any dependence of the blockage with respect to the size of the domain (regardless of the fact that the boundary conditions at the sides are periodic)?

We have not tested this for this setup. But we have carried out tests for different domain length (Lx) and width (Ly, periodic) for smaller wind farms and domains. The effect on the wake flow and the flow blockage was negligible.

L168, it's not totally clear how the height limitation set to the recycling of turbulence helps to avoid the increase in the BL height, is the subsidence mentioned in L90 not sufficient? However, it is clear that this region represents only a small portion of the total domain length, albeit still large (10 km).

We think that this sentence is clear. If no turbulence is recycled to the inflow above a certain height, then the flow is laminar above that height and thus the (turbulent) BL is limited to that height. In the CBL- case the BL growth is compensated by the subsidence. However, the BL could potentially slightly grow before the recycling plane due to the blockage effect of the wind farms. This growth would be mapped to the inflow and reinforce itself without the above mentioned recycling limit.

L194, please clarify the usage of the concept "steady state". It seems that in some instances the authors use these words when referring to a flow that has achieved statistical convergence such as in a developed BL which could be confusing with a flow that is indeed steady-state and as such, lacks transient features such as those reproduced by LES.

Also due to a comment of the other reviewer we rephrased and added information:

"While steady-state turbulence is reached after only a few hours, achieving a steady-state mean flow can take several days. Here, we declare the mean flow as steady if the oscillation amplitude of the hub height mean wind speed is less than 0.5 % and declare the turbulence as steady if the change in friction velocity is less than 2 % in 4 h."

L230,235, could the issues pointed at by Basu et al. (2008), such as those related to the estimation of the friction velocity, be added to these arguments?

If we understood it correctly, then the issues pointed out by Basu et al. (2008) are only valid for SBLs.  This section (L230 ff) deals with a CBL.

L378, is it possible to provide an example (or a threshold) of a park size where, following these arguments, the turning would occur in the clockwise direction?

We think that deriving such a rule is not possible without performing additional simulations. It might also depend on stability and BL height, because the clockwise turning is caused by wind veer. Respective studies will be addressed in a follow up study.

In Figs 5 and 6, scale is inverse with respect to the other (red to green and green to red), please maintain consistency to denote low to large values in scaling. Furthermore, it is rather common to see blue to red to depict small to large values (e.g. subfigure (f) in figures 4, 6 and 7), so to use blue in the middle can be somewhat disconcerting. Is this perhaps a choice to highlight the wakes behind the clusters?

This choice is intentional. Blue and especially red should indicate the wake, which is smaller wind speed but higher TI. We put blue in the middle to avoid a green-red color transition which might be difficult to see for red-green colorblind people.

L386, the TKE does consider the subgrid scales. Since the resolution is relatively coarse to re-solve the flow around the rotors, a sizable part of the TKE is potentially found within the subgrid scales. Please provide an argument regarding why only resolved scales should be considered in this analysis.

Resolved and SGS-TKE is displayed in the following figures and justifies that the SGS-TKE has no significant contribution here.

[Figure]

TKE and SGS-TKE at hub height in the wake 5D behind the turbine

In line 387 (equation for TI and TKE) we added:

"The SGS-TKE is neglected, because it is smaller than 10 % of the resolved TKE, inside the wind turbine wakes at a distance of 3D or more."

Figure 7, label in vertical scale should be z.

We corrected that.

L505, how is the power computed for each turbine?

The power is part of the output quantities of the advanced actuator disc model. We added a reference to an article of Wu and Porté-Agel (2010), which describes details of this model (in English language) in line 94 so that the interested reader can refer to this article.

DOI: 10.1007/s10546-010-9569-x

L608, it is remarked that for the SBL-300-7D case, the W gpg,wt component is dominant but in the

Fig. 10(e) it seems that W vke f is about the same value throughout except only within the two gaps between farms.

Yes that is true. We rephrased that to:

"For the first 10 km of the wind farms the vertical kinetic energy flux dominates but further downstream, the energy input by the pressure gradient below rotor top level is greater or equal to the vertical kinetic energy flux"

Technical corrections

• L121, typo, replace "can not" for "cannot"

- L162: I suggest to remove first comma

- L308, typo, it should be "effect" (it says affect)

- L637, typo, "of" instead of "if"

We thank for these hints, and have considered them.

---

## Referee Report (RR1)

I thank the authors for their careful consideration of my comments and suggestions. Many of my comments have been addressed very effectively, and I believe this has improved the quality manuscript. However, I feel that two of my main concerns have not been fully resolved yet, and I believe that these aspects need more attention. I clarified below why I think my main concerns are not sufficiently dealt with, and I also listed some other minor comments. I hope the authors will take the time to resolve these remaining concerns.

Remaining main concerns:
- In response to my original comment on including analyses of the smaller wind farms (main concern #2), the authors stated that additional discussions would lead to a too long paper, and as a compromise they added one additional figure. The authors also stated that more detailed investigations of the differences between small and large wind farms is subject to future work. I agree with the authors that the paper is already quite long and adding 4 more figures would probably make the paper too long. However, I am not satisfied with the current solution of adding one figure and a paragraph somewhere in the middle of the paper, and the reason is twofold. First, the narrative of the paper and hence the expectations of the reader have not changed in this revised manuscript, so one of the main messages of the paper is still focused on the difference in flow behavior of small versus very large wind farm clusters. For instance, the revised manuscript still contains several statements introducing or summarizing these differences as one of the key findings:
    - line 6: "the results show that very large wind farms cause flow effects that small wind farms do not";
    - line 681: "These results show that the power output and the wake of very large wind farms behave very differently compared to small wind farms";
    - line 711: "Overall, the results show that very large wind farms trigger much more complex flow effects than small wind farms do."

  Second, if you are not analyzing the LES data in detail to show the differences between small and large wind farm clusters, then why did you run such a large domain? You could also just run an LES of zones 2 and 3 (as a matter of fact, for the current wind direction there seems to be no interaction between zones 1 and 2+3, so you could also run separate LES of zone 1 and zone 2+3. Is there any benefit of running one big simulation?). Knowing that you ran these massive large-eddy simulations including smaller wind farm clusters (so the data is there), I still wonder while reading the paper why the comparison between small and large wind farms is not conducted consistently for every aspect analyzed throughout the paper. I think you can take two approaches here. One approach would be to do every analysis consistently for small and large cluster, and try to reduce the length of the paper by condensing the figures and make more optimal use of space (for example, figures 7 and 8 take up two entire pages, but do you really need a color figure per case? Do you fully discuss the shown velocity contours of all cases, or could you replace this with a figure showing the IBL growth for various cases?). This would be the preferred approach from a scientific point of view, but it might require redesigning some of the figures. The second approach is what you intended to do, i.e., refer more detailed investigations to future work. However, in this case I think you need to manage the reader's expectations better and say up front what the main focus of the paper is (flow behavior in very large wind farms). Moreover, you should indicate to what extent you will address the

differences with smaller wind farms, and mention in the paper when certain investigations are out-of-scope but will be part of a follow up study.

- Considering my original main concern #3, I appreciate that the authors included the perturbation pressure gradient in the energy analysis. However, I still have a couple of issues with the energy budget analysis:
  - I feel that the divergence of horizontal advection of kinetic energy is an equally important term of the energy budget equation: it is effectively a source of energy (i.e., a positive term in the budget equation) which will be significant near the wind farm leading edge, and it also shows how the kinetic energy below rotor top level is depleted by the wind farm (for example, it explains why you see a wind farm wake). I think that this term is essential for energy budget analysis of finite wind farms, and I believe that it should therefore be included in the analysis and in the figures. Now it is only mentioned as a side note to explain a difference with literature, but I don't think that is sufficient.
  - I still have an issue with the term "total energy input". I appreciate that the authors now include the perturbation pressure, but as I said before the divergence of horizontal advection is also an energy source (a positive term in the budget equation). You could argue that vertical flux and pressure gradients are external sources adding energy to the region below rotor top level and are therefore called the total energy input, but then you should specify that that is how you define an energy input. In that case, however, depletion of the kinetic energy flux is another mechanism that needs to be considered. Saying that wind turbines extract x% of the total energy input without accounting for how much they deplete the energy flux below the rotor top level is only telling half of the story.
  - Note that line 617-618 still mentions the total energy input as W_vkef+W_gpg,wt. Maybe add an equation that defines the total energy input W_total,wt.
  - In your reply to my main concern 3 subquestion b, you say that the power input by the perturbation pressure averaged over the entire farm length is approximately zero, and this justifies the 70% statement. This makes the paragraph from line 618 to 625 starting with "The wind turbines extract approximately 70 % of the total energy input …" even more confusing to me. Where does this 70% hold? Is this averaged over the entire farm length, or is it only in the bulk of the farm, or does it hold everywhere? Further, I don't understand the note you added on the divergence of the horizontal advection. You report a higher percentage of the total energy input extracted by the wind turbines than the reference values for infinite wind farms, so how can additional energy from divergence explain that you already extract more energy?

Minor comments:
- Line 581: The revised manuscript still says "… can be compensated for by two sources of energy:". This should be "three sources" (or four if you decide to include the divergence as a possible source of energy).
- Related to my concern about the clockwise flow deflection above the boundary layer: It is good to hear that new investigations show that the clockwise flow deflection

remains with the damping layer farther away. However, nothing is changed in the paper, so other readers might still be faced with the same doubts. Please add the fact that new investigations support the validity of the results to the paper where appropriate.

- Caption of figure 11: Specify what is indicated by the yellow regions.
- Section 2.4 consists of one very long paragraph which makes it difficult to read. Split up into several paragraphs for readability.
- Line 348: You are mixing up subcritical and supercritical. Supercritical means Fr>1, subcritical means Fr<1.
- Line 678-679: "This redistribution is done by a favorable perturbation pressure gradient … (not shown in Fig. 11)." You added the perturbation pressure gradient work in the revised figure, so I guess the statement between brackets can be removed?

---

## Author Response (AR2)

Response to reviewer 1:

We thank the reviewer for his additional comments and we highly appreciate his support for further improving the manuscript. We marked the reviewers comments blue and our answers black.

**Main concern #2:**
In response to my original comment on including analyses of the smaller wind farms (main concern #2), the authors stated that additional discussions would lead to a too long paper, and as a compromise they added one additional figure. The authors also stated that more detailed investigations of the differences between small and large wind farms is subject to future work. I agree with the authors that the paper is already quite long and adding 4 more figures would probably make the paper too long. However, I am not satisfied with the current solution of adding one figure and a paragraph somewhere in the middle of the paper, and the reason is twofold. First, the narrative of the paper and hence the expectations of the reader have not changed in this revised manuscript, so one of the main messages of the paper is still focused on the difference in flow behavior of small versus very large wind farm clusters. For instance, the revised manuscript still contains several statements introducing or summarizing these differences as one of the key findings:
o line 6: "the results show that very large wind farms cause flow effects that
        small wind farms do not";
o line 681: "These results show that the power output and the wake of very large
        wind farms behave very differently compared to small wind farms";
o line 711: "Overall, the results show that very large wind farms trigger much
        more complex flow effects than small wind farms do."

We think that the abstract and introduction clearly state, that the main focus of the paper is on very large wind farms and that hence the expectations of the reader are directed in the right direction. E.g. in the abstract we write:
> "The objective of this large-eddy-simulation study is to investigate the wake properties and the power output of very large potential wind farms in the German Bight [...]"

In the introduction we write:
> *"We provide new insights into the wake properties and power output of very large wind farms and how these depend on the varied parameters. Specifically we want to answer these questions:*
> *1. How is the flow inside and above the boundary layer affected by very large wind farms?*
> *[...]*
> *4. How much power output or power density can be expected for very large wind farms?*
> *[...]"*

However, it is true that we make several comparisons to small wind farms, also in our final and main conclusions. We do that to highlight the characteristics of very large wind farms, which would be difficult without a comparison to small wind farms. We do not investigate the small wind farms in the same detail as we do for the large wind farms, because many other papers already have investigated small wind farms (as we have shown in the introduction). We find that it is sufficient to refer to and cite the findings of other authors about small wind farms.

Second, if you are not analyzing the LES data in detail to show the differences between small and large wind farm clusters, then why did you run such a large domain? You could also just run an LES of zones 2 and 3 (as a matter of fact, for the current wind

direction there seems to be no interaction between zones 1 and 2+3, so you could also run separate LES of zone 1 and zone 2+3. Is there any benefit of running one big simulation?). Knowing that you ran these massive large-eddy simulations including smaller wind farm clusters (so the data is there), I still wonder while reading the paper why the comparison between small and large wind farms is not conducted consistently for every aspect analyzed throughout the paper.

We agree that two more idealized and/or smaller setups, one for small and one for large wind farms would enable us to make better comparisons between small and large wind farms. We have chosen this special setup including large and small wind farms, because of two reasons:

1. We want to show the wake effects and power output for the specific case of the German 2040 expansion target in the German Bight for the most typical weather situations. We think that the investigation of this special case is of great interest for researchers and industry involved in the energy transition and for the society in general.

2. It was a constraint by the project funder to include all wind farms in all priority areas in the German Exclusive Economic Zone.

 I think you can take two approaches here.
One approach would be to do every analysis consistently for small and large cluster, and try to reduce the length of the paper by condensing the figures and make more optimal use of space (for example, figures 7 and 8 take up two entire pages, but do you really need a color figure per case? Do you fully discuss the shown velocity contours of all cases, or could you replace this with a figure showing the IBL growth for various cases?). This would be the preferred approach from a scientific point of view, but it might require redesigning some of the figures.
The second approach is what you intended to do, i.e., refer more detailed investigations to future work. However, in this case I think you need to manage the reader's expectations better and say up front what the main focus of the paper is (flow behavior in very large wind farms). Moreover, you should indicate to what extent you will address the differences with smaller wind farms, and mention in the paper when certain investigations are out-of-scope but will be part of a follow up study.

We decided to leave the focus of this article on very large wind farms and to not make a systematic comparison between small and large wind farms. Hence we choose approach 2. As written above, we think that introduction and abstract make already clear, that the focus lies on very large wind farms. However, we would like to follow your suggestion and tried to add statements that clarify that we do not present a systematic comparison between small and large wind farms. We found that such a statement only makes sense right at the beginning of the results section (line 300). There we added:

> To highlight the characteristics of very large wind farms some comparisons to small wind farms are made. However, the focus of this work lies on very large wind farms, so that a systematic comparison between large and small wind farms is not conducted here but will be part of a follow-up study.

• Considering my original main concern #3, I appreciate that the authors included the perturbation pressure gradient in the energy analysis. However, I still have a couple of issues with the energy budget analysis:

o I feel that the divergence of horizontal advection of kinetic energy is an equally important term of the energy budget equation: it is effectively a source of energy (i.e., a positive term in the budget equation) which will be significant near the wind farm leading edge, and it also shows how the kinetic energy below rotor top level is depleted by the wind farm (for example, it explains why you see a wind farm wake). I think that this term is essential for energy budget analysis of finite wind farms, and I believe that it should therefore be included in the analysis and in the figures. Now it is only mentioned as a side note to explain a difference with literature, but I don't think that is sufficient.

o I still have an issue with the term "total energy input". I appreciate that the authors now include the perturbation pressure, but as I said before the divergence of horizontal advection is also an energy source (a positive term in the budget equation). You could argue that vertical flux and pressure gradients are external sources adding energy to the region below rotor top level and are therefore called the total energy input, but then you should specify that that is how you define an energy input. In that case, however, depletion of the kinetic energy flux is another mechanism that needs to be considered. Saying that wind turbines extract x% of the total energy input without accounting for how much they deplete the energy flux below the rotor top level is only telling half of the story.

We agree that the advection of kinetic energy is an important term in the energy budget analysis of a wind farm. However, the intention of this analysis was rather to examine those processes that drive the wake recovery and limit the power output further inside very large wind farms.
We also agree that we have not clearly defined what we mean by "energy input", so we have rephrased the beginning of section 3.2.2:

> To examine the dependency of the wind farm efficiency on the turbine spacing and the BL height in more detail, an energy source analysis is made in this section. Here, an energy source is defined as an energy input to the flow, i.e. a process that drives the wake recovery. This can be one of the following:
>
> > 1. Vertical turbulent flux of kinetic energy at rotor top level, $W_{vkef}$
> > 2. Work done by the geostrophic pressure gradient on the flow below rotor top level (bottom of the BL), $W_{gpg,wt}$
> > 3. Work done by the perturbation pressure gradient on the flow below rotor top level, $W_{ppg,wt}$
>
> The analysis is a simplified version of the analyses made by \citet{Abkar2014} and \citet{Allaerts2017a} and does not claim to be a complete energy budget analysis. The intention of this analysis is to show which processes dominate the wake recovery and thus limit the achievable power density of very large wind farms. Thus the advection of upstream kinetic energy is not considered here. The above named sources are calculated as follows:

We also considered to include the advection term into Figure 11. Unfortunately this would require at least a doubling of the range of the vertical axis so that the entire figure becomes unreadable. So, finally we decided not to include the advection of kinetic energy in the Figure and to only mention in the text that it is a dominant energy source for the wind turbines (line 605):

The dominant energy source for the first turbine rows is the advection of kinetic energy. The advection is not included in Fig.~\ref{fig.xz_2x3_power_densities} because it is larger than the other terms and would make the quantification of the smaller terms difficult.

We renamed the section 3.2.2 from "Energy flux analysis" to the more suitable name "Energy source analysis".

We have also made some other small corrections in this section, e.g. renaming "kinetic energy flux" to "vertical kinetic energy flux". The changes can be seen in the attached "diff"-document with highlighted changes.

o Note that line 617-618 still mentions the total energy input as $W\_vkef+W\_gpg,wt$. Maybe add an equation that defines the total energy input $W\_total,wt$.

We have now done this in line 612.

o In your reply to my main concern 3 subquestion b, you say that the power input by the perturbation pressure averaged over the entire farm length is approximately zero, and this justifies the 70% statement. This makes the paragraph from line 618 to 625 starting with "The wind turbines extract approximately 70 % of the total energy input ..." even more confusing to me. Where does this 70% hold? Is this averaged over the entire farm length, or is it only in the bulk of the farm, or does it hold everywhere?

We have now specified where this 70 % hold:

The work done by the geostrophic pressure gradient on the flow below the rotor top level achieves a power density of approximately 0.6 W m$^{-2}$. It is thus not the dominating energy source inside the wind farms but it still contributes approximately 20 % to the sum of all sources $W_{total}=W_{vkef}+W_{gpg,wt}+W_{ppg,wt}$. In the downstream half of the wind farms the ratio between the wind turbine power and $W_{total}$ is approximately 70 %

Further, I don't understand the note you added on the divergence of the horizontal advection. You report a higher percentage of the total energy input extracted by the wind turbines than the reference values for infinite wind farms, so how can additional energy from divergence explain that you already extract more energy?

We deleted these sentences, because a comparison between our non-idalized and the named idealized infinite wind farm setup does not make sense.

Minor comments:
• Line 581: The revised manuscript still says "... can be compensated for by two sources of energy:". This should be "three sources" (or four if you decide to include the divergence as a possible source of energy).

We have rephrased this part, see further above.

• Related to my concern about the clockwise flow deflection above the boundary layer: It is good to hear that new investigations show that the clockwise flow deflection remains with the damping layer farther away. However, nothing is changed in the paper, so other readers might still be faced with the same doubts. Please add the fact

that new investigations support the validity of the results to the paper where
appropriate.

We have added a note on that in line 521:
> However, currently running investigations with much higher Rayleigh damping heights
> show the same behaviour.

Caption of figure 11: Specify what is indicated by the yellow regions.

Done.

Section 2.4 consists of one very long paragraph which makes it difficult to read. Split
up into several paragraphs for readability.

Done.

Line 348: You are mixing up subcritical and supercritical. Supercritical means Fr>1,
subcritical means Fr<1.

We found a mistake in line 342, where Fr<1 (smaller than 1) was accidently stated as supercritical.
We now corrected to Fr>1 (greater than 1).

Line 678-679: "This redistribution is done by a favorable perturbation pressure
gradient ... (not shown in Fig. 11)." You added the perturbation pressure gradient work
in the revised figure, so I guess the statement between brackets can be removed?

Yes, we have removed it.

Response to reviewer 2:

We thank the reviewer for his additional comment. We marked the reviewers comment blue and our answer black.

Thank you for your re-submission. You have addressed essentially all the comments to my first review by making adjustments to the manuscript.

I point now only to one technical detail:
Please only note that regarding the previous comment "L505, how is the power computed for each turbine" you have included a reference, Wu and Porté-Agel. Please check if the year is correct as the work you refer seem to be from 2011 instead of 2010. However, if that was the case (it does when comparing the DOI you provide), the power calculation seem to be missing in that article.

The correct year is 2011. We corrected that. We also added further information about the thrust and power calculation in line 97 because it is not included in the citet references, as the reviewer has stated.
* * *
tum sink and an angular momentum source (inducing wake rotation). The ADM-R is described in detail by Steinfeld et al. (2015) and Wu and Porté-Agel (2011). The actuator disc is divided into several segments along the radial and tangential direction to allow for a non-uniform thrust distribution over the disc. The lift and thrust force of each segment $f_l$ and $f_d$ is projected
* * *
on the axial ($f_a$) and tangential ($f_t$) direction:

100
$$f_a = -f_l \cos \Phi - f_d \sin \Phi \,, \qquad\qquad f_t = -f_l \sin \Phi - f_d \cos \Phi \,, \qquad\qquad (4)$$

where $\Phi$ is the angle between the local wind vector and the disc. The rotor thrust $F$ and torque $M$ are then calculated as the sum over all $N$ segments at radius $r_i$:

$$F = \sum_{i=1}^{N} f_{a,i} \,, \qquad\qquad M = \sum_{i=1}^{N} f_{t,i} r_i \,. \qquad\qquad (5)$$

The wind turbine power is calculated out of the rotational speed of the rotor $n_{rotor}$ and the torque:

105
$$P = 2\pi \, n_{rotor} \, Q \qquad\qquad (6)$$